# A comprehensive platform for analyzing longitudinal multi-omics data

Suhas V. Vasaikar [1] ✉, Adam K. Savage [1], Qiuyu Gong[1], Elliott Swanson [1,2], Aarthi Talla[1], Cara Lord[1,3], Alexander T. Heubeck [1], Julian Reading [1], Lucas T. Graybuck [1], Paul Meijer [1], Troy R. Torgerson[1], Peter J. Skene[1], Thomas F. Bumol [1] & Xiao-jun Li [1] ✉

Longitudinal bulk and single-cell omics data is increasingly generated for biological and clinical research but is challenging to analyze due to its many intrinsic types of variations. We present PALMO (https://github.com/aifimmunology/PALMO), a platform that contains five analytical modules to examine longitudinal bulk and single-cell multi-omics data from multiple perspectives, including decomposition of sources of variations within the data, collection of stable or variable features across timepoints and participants, identification of up- or down-regulated markers across timepoints of individual participants, and investigation on samples of same participants for possible outlier events. We have tested PALMO performance on a complex longitudinal multi-omics dataset of five data modalities on the same samples and six external datasets of diverse background. Both PALMO and our longitudinal multi-omics dataset can be valuable resources to the scientific community.

Applying multi-omics technologies to measure longitudinal specimens of human participants provides unprecedented insights on disease such as COVID-19[1–3], diabetes[4] and lymphoma[5]. Single-cell technologies such as single-cell ribonucleic acid sequencing (scRNA-seq) and single-cell assay for transposase-accessible chromatin sequencing (scATAC-seq) can offer granular details on disease mechanisms and are increasingly utilized in biological and clinical research[6–8]. It is anticipated that more and more longitudinal bulk and single-cell omics data will be generated by the scientific community.

Different statistical methods are used to analyze longitudinal data to account for the diversities in research interest, study design, and/or data type (continuous or categorical)[9,10]. Generalized linear mixed model (GLMM) is a popular approach for analyzing continuous longitudinal data. It is common that the same dataset can be examined from multiple perspectives with different methods. Complications such as human heterogeneity, interdependency between multiple samples of same participant, missing and/or incomplete data,

unbalanced dataset, and unexpected outlier events (e.g., severe adverse events in clinical trials) are all intrinsic to longitudinal data. The usage of single-cell technologies brings additional complications such as dropout, sparseness, interdependency between cells of same sample, and unbalanced cell counts in individual samples[11,12]. Advanced methods have been applied to analyze longitudinal bulk omics data with customized codes for specific projects[4,13]. Sophisticated methods for analyzing cross-sectional single-cell omics data have also been developed with mixed performance[14–18]. While software tools such as variancePartition[19] and tcR[20] can be repurposed to examine longitudinal omics data either from a single perspective and/or collected on a single technical platform, we are not aware of any well-accepted software package that is specifically designed to analyze longitudinal bulk and single-cell omics data. Instead, researchers rely on customized codes to analyze such data, which is time-consuming, error-prone and a non-small challenge to many people. A comprehensive yet simple-to-use software tool to extract insightful information from longitudinal omics data is desired.

[1]Allen Institute for Immunology, Seattle, WA 98109, USA. [2]Present address: Department of Genome Sciences, University of Washington School of Medicine, Seattle, WA, USA. [3]Present address: GlaxoSmithKline, Collegeville, PA 19426, USA. ✉e-mail: suhas.vasaikar@alleninstitute.org; xiaojun.li@alleninstitute.org

Here, we present PALMO (https://github.com/aifimmunology/PALMO), a software package designed to analyze longitudinal bulk and single-cell omics data (Fig. 1a). Five analytical modules are implemented in PALMO (Fig. 1b): (i) Variance decomposition analysis (VDA) evaluates contributions of factors of interest to the total variance of individual features (Fig. 1c). (ii) Coefficient of variation (CV) profiling (CVP) assesses intra-participant variation over time in bulk data and identifies consistently stable or variable features among participants (Fig. 1d). (iii) Stability pattern evaluation across cell types (SPECT) assesses longitudinal stability patterns of features in single-cell omics data and identifies stable or variable features that are unique to individual cell types but consistent among participants (Fig. 1e). (iv) Outlier detection analysis (ODA) examines the possibility of abnormal events occurring during a longitudinal study (Fig. 1f). (v) Time course analysis (TCA) evaluates transcriptomic changes over time based on longitudinal scRNA-seq data of the same participant and identifies genes that exhibit significant temporal changes (Fig. 1g). Together these five modules provide unique insights on longitudinal omics data from multiple perspectives. We also developed functions to display CVs of features of interest in circos plots (Fig. 1h). We test PALMO performance on a complex longitudinal multi-omics dataset of five data modalities and six external datasets of diverse background.

## Results

### A complex longitudinal multi-omics dataset to demonstrate PALMO performance

To demonstrate PALMO performance, we collected sixty blood samples (plasma and peripheral blood mononuclear cells (PBMCs)) from six healthy, non-smoking Caucasian donors (three females and three males) between 25 to 38 years old over a 10-week period (Supplementary Fig. 1a). Complete blood count (CBC) was collected on all these samples (Supplementary Data 1a). The abundance of 1,156 plasma proteins were measured on these samples as well (Supplementary Data 1c), but only 1,042 (68%) proteins had reliable quantification results (Supplementary Data 2a). High-dimensional flow cytometry and droplet-based scRNA-seq assays were performed on a subset of 24 PBMC samples from four donors (one female and three males) over Week 2 to 7. A total of 27 cell types were identified from flow cytometry data (Supplementary Fig. 2, Supplementary Data 1b). Droplet-based scATAC-seq assay was also performed on 18 out of the 24 PBMC samples. This multi-omics dataset of five data modalities on the same samples can be a valuable resource to the scientific community for immune health study and/or software development.

We retrieved high quality scRNA-seq data of 472,464 cells and labeled them to 31 different cell types using Seurat level2 labelling[16] (Supplementary Fig. 3a, b, Supplementary Data 3a). Among the nineteen overlapping cell types identified by both scRNA-seq and flow cytometry, the corresponding cell frequencies as measured by the two data modalities were highly correlated (two-sided $p < 0.05$ on Pearson correlation as evaluated by R function "cor.test()") except for those of double negative T (dnT) cells (Supplementary Fig. 3c). Unless specified otherwise, we filtered out low frequent cell types (average frequency <0.5%) and kept 19 out of the 31 cell types for downstream analysis (Supplementary Data 3b). We also kept only 11,191 genes that had an average (across timepoints) expression of 0.1 or higher in at least one cell type of one donor.

scATAC-seq data was analyzed using the ArchR[21] package. We observed 294,623 peaks in 135,566 cells after removing doublets. Cells were labeled to 28 different cell types using genescore matrix as implemented in ArchR (Supplementary Fig. 3d, e). We noticed the

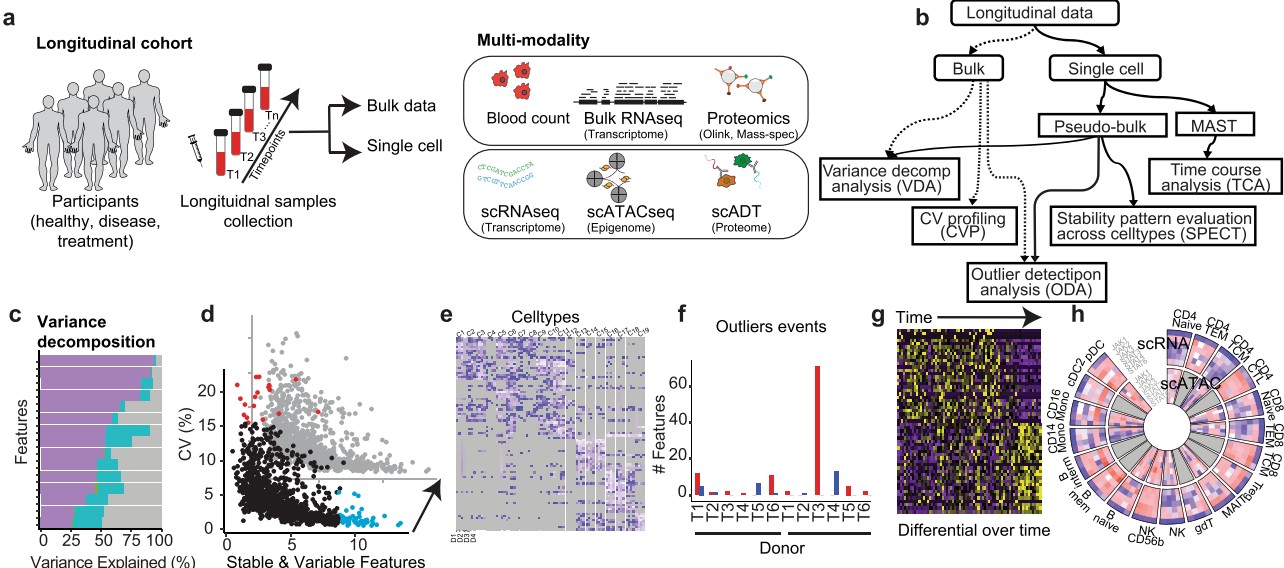

**Fig. 1 | General workflow and analysis schema of PALMO. a** PALMO can work with complex longitudinal data, including clinical data, bulk omics data, and single-cell omics data. **b** Overview of five analytical modules implemented in PALMO.
**c** Variance decomposition analysis (VDA) applies generalized linear mixed model to assess contributions of factors of interest (such as disease status, sex, individual participant, cell type, experimental batch, etc.) to the total variance of individual features in the data. **d** Coefficient of variation (CV) profiling (CVP) is designed for bulk longitudinal data, calculates CV of repeated measurements on the same participant to assess the corresponding longitudinal stability, and compares CVs of different participants to identify consistently stable or variable features. **e** Stability pattern evaluation across cell types (SPECT) is the CVP counterpart for single-cell omics data, analyzes stability patterns of features across different cell types and different participants, classifies features based on how often they are stable or variable in cell type-donor combinations, and identifies features that are unique to individual cell types and consistent among participants. **f** Outlier detection analysis (ODA) evaluates how many features in a sample are outliers when compared with the corresponding features in other samples of same participant, assesses whether the number of outlier features in the sample is significantly higher than expectation, and identifies possible abnormal events occurred during a longitudinal study. **g** Time course analysis (TCA) uses the hurdle model to evaluate transcriptomic changes over time based on longitudinal scRNA-seq data of same participants, models time as a continuous variable for data with at least three timepoints, and identifies up- or down-regulated genes over time. **h** PALMO uses circos plots to display CVs of features of interest and reveal stability patterns across features, participants, cell types, and data modalities. Adobe Illustrator (version 27.1.1; https://www.adobe.com/products/illustrator.html) was used to draw (**a**), arrange panels, and edit text. PowerPoint (version 16.69; https://www.microsoft.com/en-us/microsoft-365/powerpoint) was used to draw (**b**).

labeling scores on scATAC-seq data were much lower than the corresponding scores on scRNA-seq data, likely reflecting the challenge in cell labeling on scATAC-seq data. We filtered out low quality cells (labeling score <0.5), removed cell types having less than 50 remaining cells, and kept 14 out of the 28 cell types for downstream analysis (Supplementary Data 3b). We also kept only 24,769 genes that had an average (across timepoints) gene score of 0.1 or higher in at least one cell type of one donor.

In addition to our own data, we also evaluated PALMO performance against six external omics datasets of diverse complexities, different sample types and/or different technical platforms (Supplementary Fig. 1b). More examples of PALMO usage beyond those presented here can be found in PALMO vignettes (https://github.com/aifimmunology/PALMO/blob/main/Vignette-PALMO.pdf), including performance on unbalanced data, data with replicates, and data of a single donor with multiple timepoints.

## Application of VDA to assess sources of variations

We applied VDA to evaluate inter- and intra-donor variations in our bulk data (CBC, PBMC frequencies from flow cytometry, and plasma proteomics data), using donor and week (timepoint) as factors of interest. CBC measurements showed strong inter-donor variations and minuscule intra-donor variations (Supplementary Fig. 4a, b). PBMC frequencies from flow cytometry showed very strong inter-donor variations (Supplementary Fig. 4c, d) with intra-class correlation (ICC) ranging from 51% (IgD CD27⁻ B cells) to 98% (CD4 Temra: CD4⁺ effector memory T cells re-expressing CD45RA). In comparison, the highest ICC on intra-donor variations was 19% (cDC1: conventional type 1 dendritic cells). Plasma proteins followed a similar trend with some exceptions (Supplementary Fig. 4e, f, Supplementary Data 2a). Inter-donor variations of 621 (60%) out of the 1042 quantified proteins contributed more than 50% to the corresponding total variance. Only 75 proteins (7%) had more intra-donor variation than inter-donor variation, but none contributed more than 50% to the total. A previous study[22] identified 155 proteins having high inter-donor variations, 81% (126) of which overlapped with the 621 inter-donor variable proteins.

We added cell type as a factor of interest in the VDA of our scRNA-seq and scATAC-seq data. Inter-cell-type variations were more prominent than inter- and intra-donor variations in both single-cell data modalities. Based on our scRNA-seq data, 10, 0, and 4384 genes had more than 50% of total variance from inter-donor, intra-donor, and inter-cell-type variations, respectively (Fig. 2a, Supplementary Data 3c). Nine of the top ten inter-cell-type variable genes (ICC: 98–99%, Fig. 2b) have known immune functions (Supplementary Data 3d). The top gene, LILRA4, is predominantly expressed in plasmacytoid dendritic cells (pDCs) and prevents pDCs from over-blown reaction to viral infections[23]. Six of the top ten inter-donor variable genes (ICC: 53–94%, Fig. 2c) are linked to the X or Y chromosome and seven of them showed differential expression between ovary and prostate/testis, reflecting the sex difference between male and female donors. Contributions from intra-donor variations to the total variance were small (ICC ≤ 3%, Fig. 2d), indicating the immune systems of the four healthy donors were quite stable over the study period.

The VDA results on our scATAC-seq data, using genescore matrix, showed similar trends as that on our scRNA-seq data (Fig. 2e). A total of 33, 0, and 7847 genes had more than 50% of total variance from inter-donor, intra-donor, and inter-cell-type variations, respectively (Supplementary Data 3e). All the top ten inter-cell-type variable genes (ICC: 95–97%, Fig. 2f) have known immune functions (Supplementary Data 3f). The top gene, SPIB, is an enhancer regulating pDC development[24]. Among the top ten inter-donor variable genes (ICC: 58–89%, Fig. 2g), XIST, ZNF705D, GTF2IRD2, and USP32P2 have differential expression between ovary and prostate/testis; RHD encodes a key protein in the Rh blood group system; and GSTM1 belongs to a highly polymorphic supergene family and affects heterogeneous response to toxicity[25]. These genes appeared to capture more diverse types of differences among donors than their counterparts from scRNA-seq data. The ICCs of the top five intra-donor variable genes (ICC: 32–34%, Fig. 2h) were about 10-fold higher than that of the corresponding top gene, JUN, by scRNA-seq data, suggesting chromatin accessibility might be more sensitive to biological changes than gene expression.

variancePartition[19] was previously developed to study variations in gene expression data and can be applied to longitudinal omics data for the same purpose. VDA generated almost identical results as variancePartition on two tested datasets after removing missing values (Supplementary Fig. 5), which was needed to run variancePartition but not VDA.

VDA can be used to study T-cell receptor (TCR) repertoires. Previously sorted CD4⁺ and CD8⁺ non-naïve T cells were isolated from PBMC samples of four systemic sclerosis (SSc) donors and analyzed to obtain sequencing data of TCR β-chains[26]. The data was originally analyzed using tcR[20], which was developed specifically for TCR data with functions either providing sample-level views on the whole repertories or treating clonotype data as binary (present or absent). We downloaded the TCRβ data (GSE156980) and calculated the frequency of unique clonotypes from both CD4⁺ and CD8⁺ T cells. A total of 288,597 unique clonotypes were obtained from CD4⁺ T cells and 11,739 from CD8⁺ T cells, respectively. We treated the clonotype data as continuous and used donor, time, and subtype (limited SSc versus diffuse SSc) as factors of interest in VDA. We identified from CD4⁺ T cells 6,625, 3, and 41 clonotypes having more than 50% of total variance from inter-donor, intra-donor, and inter-subtype variations, respectively (Supplementary Fig. 6a–d, Supplementary Data 4a). The corresponding counts from CD8⁺ T cells were 650, 0, and 1 (Supplementary Fig. 6e–h, Supplementary Data 4b). As illustrated in Supplementary Fig. 6b, f, many inter-donor variable clonotypes were donor-specific and stable over time, making them potential candidates responsible for SSc pathogenesis. The identification of inter-subtype variable clonotypes (Supplementary Fig. 6d, h) is interesting since some of them might be specific to either limited SSc or diffuse SSc. VDA provided additional insights on the TCR data beyond the original study[26].

## Application of CVP to evaluate longitudinal stability

We applied CVP to identify longitudinally stable and variable proteins from our proteomics data (Fig. 3a). The distribution of median CV (among donors) peaked near 5% (Supplementary Fig. 7a), which we used as a cut-off to separate variable (median CV > 5%) and stable (median CV < 5%) proteins (Supplementary Data 2b–d). A total of 413 proteins were longitudinally variable, among which SNAP23, GRAP2, ARG1, AIFM1, and MESD had the highest median CV (24.6-27.7%, Fig. 3b). Such moderate CV values are consistent with the observed low intra-donor variations by VDA. A total of 629 proteins were longitudinally stable, among which SOD2, NRP2, OSCAR, NRCAM, and MIA had the lowest median CV (0.6-0.8%, Fig. 3c). These stable proteins may be interesting biomarker candidates if they change under some disease conditions. They can also be used to bridge proteomics data of different experimental batches.

## Application of ODA to discover a possible abnormal event

We noticed that proteomics data of donor PTID3 exhibited higher CV values than those of other donors (Fig. 3a) and weaker intra-donor correlations at week 6 than at other weeks (Supplementary Fig. 7b). We applied ODA to check whether donor PTID3 had an abnormal event at week 6. We selected |z|>2.5 as the criterion for outliers so that just above 1% of all quantifiable proteins are expected to be outliers. More accurately, we expected 1.24% of proteins, i.e., 19 proteins per donor per time point, to be outliers by chance (Methods). A total of 71 outlier

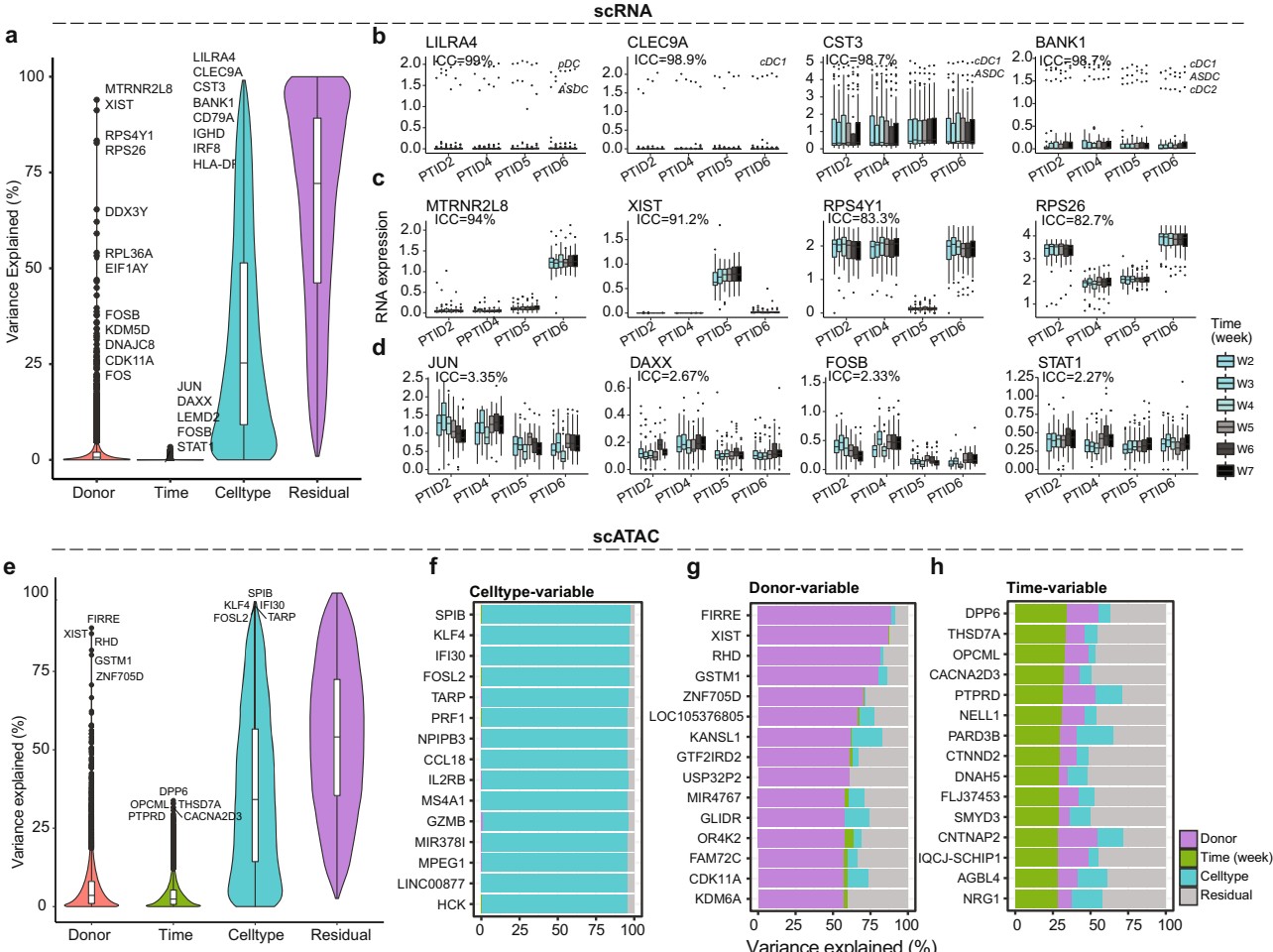

**Fig. 2 | Variance decomposition on longitudinal single-cell omics data. a** Overall distributions of variance explained by inter-donor variations (Donor), longitudinal intra-donor variations (Week), variations among cell types (Celltype), or residual variations (Residual) based on scRNA-seq data. The scRNA-seq data was collected on 24 independent peripheral blood mononuclear cell (PBMC) samples from $n = 4$ healthy participants with each participant contributing one sample a week for 6 weeks. The distributions were evaluated based on pseudo-bulk intensities of $n = 11,191$ genes in 19 cell types. **b, c** Examples of genes whose total expression variance was most explained by inter-cell-type variations (**b**) or inter-donor variations (**c**). **d** Examples of genes that had the most but still minuscular intra-donor variations in expression. **b–d** Pseudo-bulk intensities of the corresponding genes in 19 cell types were displayed in boxplots. **e** Same as (**a**) but based on scATAC-seq

data from $n = 18$ out of the 24 PBMC samples with 2 participants contributing 6 samples while other 2 participants contributing 3 samples. The distributions were evaluated based on gene scores of $n = 24,769$ genes in 14 cell types. **f, g** The top list of genes whose inter-cell-type (**f**) or inter-donor (**g**) variations contributed most to the total variance in scATAC-seq data. **h** The top list of genes that had the most intra-donor variations in scATAC-seq data. **a–e** Each boxplot displays the median (centerline), the first and third quartiles (the lower and upper bound of the box), and the 1.5x interquartile range (whiskers) of the data. ICC: intra-class correlation. Adobe Illustrator (version 27.1.1; https://www.adobe.com/products/illustrator. html) was used to arrange panels and edit text. Source data are provided as a Source Data file.

proteins were identified at Week 6 on donor PTID3 (adjusted $p = 2.7 \times 10^{-26}$, Fig. 3d, e, Supplementary Data 2e, f). Eight of the top ten proteins having the highest $z$ scores (2.84–2.85) play important roles in immune response and immunity (Supplementary Data 2g). Gene set enrichment analysis (GSEA) revealed the outlier proteins were enriched in immunological processes such as adaptive immune responses, antigen processing and presentation via major histocompatibility complex (MHC) class II, T cell activation, etc. (Supplementary Fig. 7c). Single-sample GSEA (ssGSEA)[27] on all PTID3 samples identified Week 6 as an outlier and revealed increased activity at Week 6 in important immune processes (Supplementary Fig. 7d), including MYC targets (v1 and v2)[28], interferon-alpha and gamma responses[29], androgen response[30], pancreas beta cells[31], and peroxisome[32]. Although further validation is required, these results suggest the possible occurrence of an immunological perturbation event (such as infection) experienced by PTID3 at week 6. Such outlier phenotypes can be obscured by analyses focusing on differences between sample groups.

**Application of SPECT to reveal diverse gene stability patterns**
We applied SPECT to analyze our scRNA-seq data. Noticing the two well-known housekeeping genes, ACTB and GAPDH, had CVs (across timepoints) just above 10% in some cell types (Supplementary Fig. 8), we used a CV cut-off of 10% to separate longitudinally variable (CV > 10%) or stable (CV < 10%) genes in individual cell types of individual donors. We then counted how many times individual genes were variable and/or stable in the 76 combinations between donor ($n = 4$) and cell type ($n = 19$). A gene was denoted as super variable (SUV) or super stable (SUS) if it was variable or stable in at least 40 donor-cell type combinations. A gene was denoted as variable across time in cell-types (VATIC) or stable across time in cell-types (STATIC) if it was variable or stable in at least one cell type across all donors but in less than 40 donor-cell type combinations. We identified a total of 700 SUV genes (Supplementary Fig. 9a), 2129 SUS genes (Supplementary Fig. 9b), 5750 VATIC genes, and 4004 STATIC genes from the dataset. Since a gene can be consistently

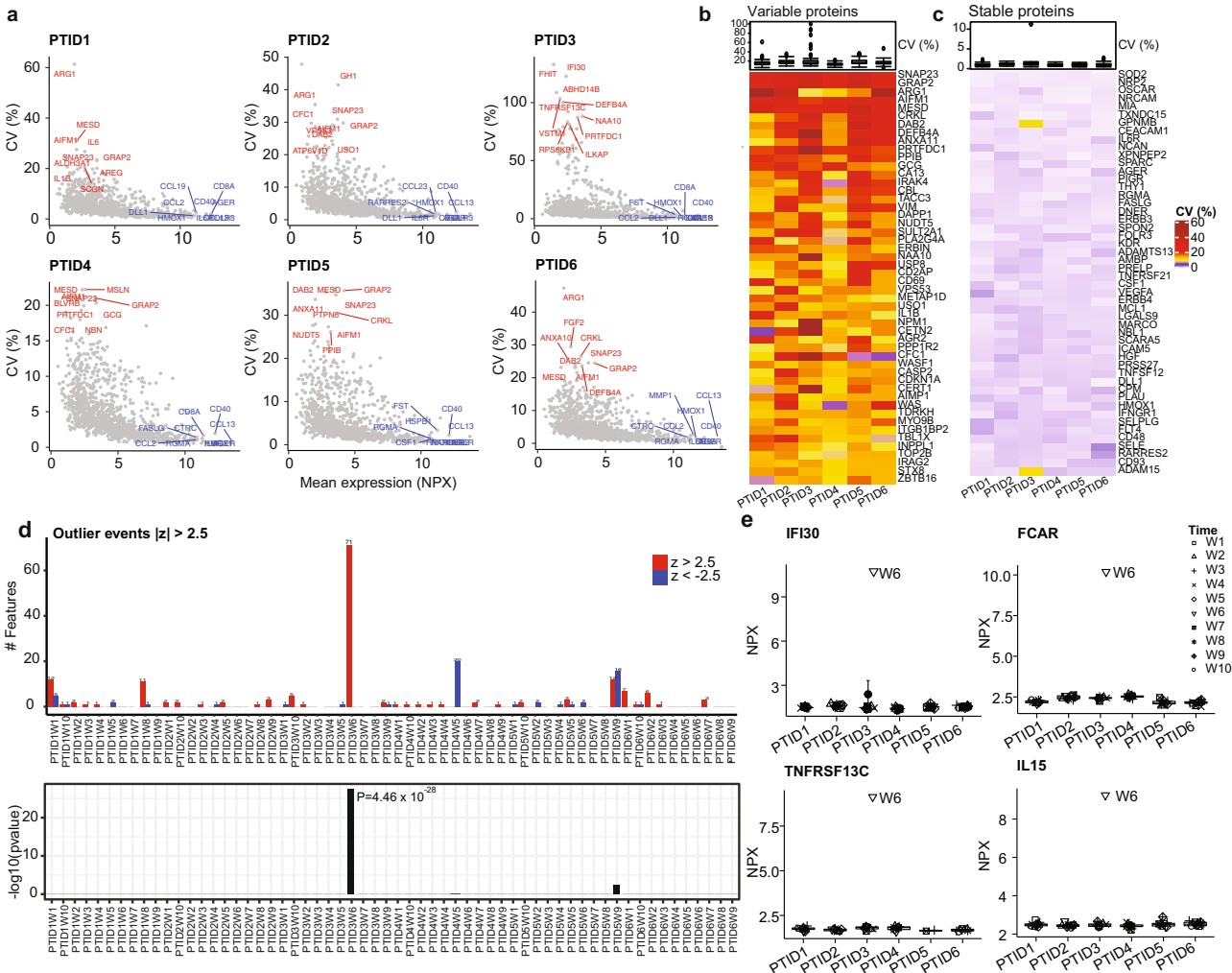

**Fig. 3 | Longitudinal stability of plasma proteome. a** Scatter plots of coefficient of variation (CV) versus mean of normalized protein expression (NPX) over 10 timepoints in $n = 6$ participants. One plasma sample per week was collected from $n = 6$ participants over 10 weeks. The evaluation for each participant was based on measurements on 1042 proteins in the corresponding 10 plasma samples. The longitudinal stable and variable proteins are represented in blue and red, respectively. **b**, **c** Heatmap of CV of top 50 longitudinally variable (**b** CV > 5%) or stable (**c** CV < 5%) plasma proteins. **d** Top panel: Number of proteins with $z > 2.5$ (red) or $z < -2.5$ (blue) in individual samples, where $z = (NPX - \overline{NPX})/SD$ with $\overline{NPX}$ and $SD$ being the mean and the standard deviation, respectively, of $NPX$ across samples of the same participant. Bottom panel: $-\log_{10}(p)$ for individual samples being possible outliers, where $p$ is calculated based on a binomial test (two-sided). **e** Protein examples clearly demonstrate that Week 6 of participant PTID3 was an outlier. **b**, **c**, **e** Each boxplot displays the median (centerline), the first and third quartiles (the lower and upper bound of the box), and the 1.5x interquartile range (whiskers) of the data. Adobe Illustrator (version 27.1.1; https://www.adobe.com/products/illustrator.html) was used to arrange panels and edit text. Source data are provided as a Source Data file.

variable in one cell type and consistently stable in another, VATIC and STATIC genes are not mutually exclusive (Supplementary Fig. 9c).

The SUV genes were enriched in 57 pathways, many of which are associated with cellular proliferation and activity (Supplementary Data 3g). Eight of the top ten SUV genes (Supplementary Data 3h) have distinct roles in gene regulation, including four transcription factors (FOS, FOSB, JUN, and KLF9), two phosphatases (DUSP1 and PPP1R15A), one regulator of mTOR pathway (DDIT4)[33], and one inhibitor of NF-κB pathway (TNFAIP3)[34]. In comparison, the SUS genes were enriched in 501 pathways of rather diverse, basic cellular processes (Supplementary Data 3i). Among the top ten SUS genes (Supplementary Data 3j), five (RPS12, RPL10, RPL13, RPLP1, and RPL41) encode ribosomal proteins and two (FTL and FTH1) encode ferritin for iron storage. Many SUS genes are more stable than ACTB and GAPDH and may be good candidates for estimating batch effects in scRNA-seq data[35].

## STATIC genes as potential biomarkers for cell types or biological conditions

We collected up to 25 top STATIC genes from each cell type and obtained 220 unique genes (Fig. 4a, Supplementary Data 5a). These 220 STATIC genes are enriched in pathways such as innate (adjusted $p = 1.43 \times 10^{-9}$) and adaptive (adjusted $p = 1.33 \times 10^{-9}$) immune response, allograft rejection (adjusted $p = 3.06 \times 10^{-16}$), lymphocyte mediated immunity (adjusted $p = 3.72 \times 10^{-8}$), myeloid mediated immunity (adjusted $p = 2.71 \times 10^{-5}$), B/T-cell proliferation (adjusted $p < 1.46 \times 10^{-3}$), acute inflammatory response (adjusted $p = 7.48 \times 10^{-3}$), hematopoietic cell lineage (adjusted $p = 2.44 \times 10^{-4}$), etc. (Supplementary Data 5b). Examples of top STATIC genes for major cell types were shown in Fig. 4b, including: GIMAP7, LEF1, CD27, CCR7, and TSHZ2 for T cells; CD79A, MS4A1, TCL1A, CD79B, and TNFRSF13C for B cells; PRF1, FGFBP2, SPON2, CST7, and KLRD1 for natural killer (NK) cells; CD14, FCN1, MNDA, SEPINA1, and SPI1 for monocytes; and LILRA4, IRF7, FCER1A, SERPINF1, and SPIB for dendritic cells (DCs). All these genes demonstrated cell

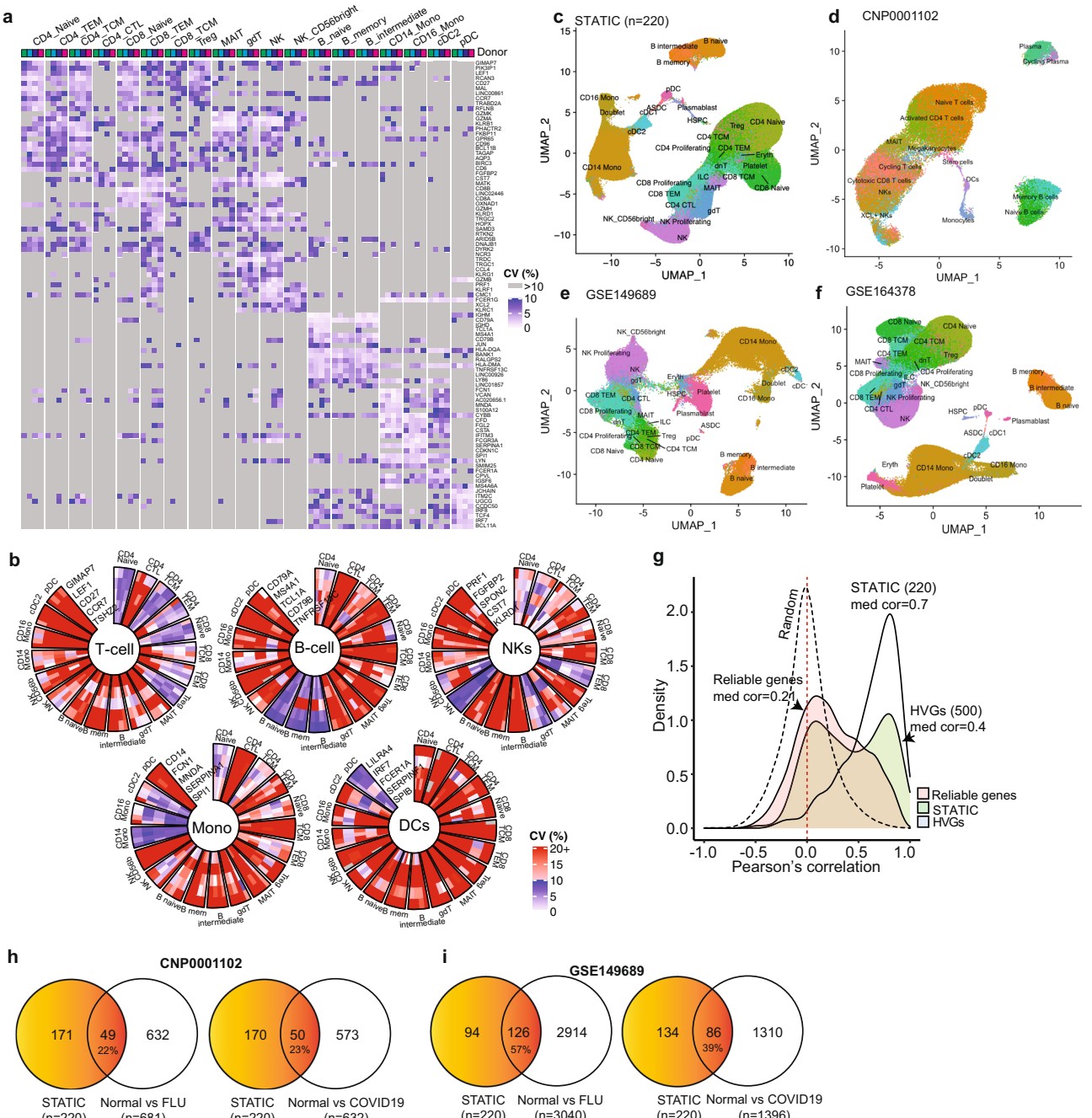

**Fig. 4 | Properties of 220 STATIC genes of PBMC. a** Heatmap of coefficient of variation (CV) evaluated on 93 out of the 220 stable across time in cell-types (STATIC) genes that were identified from 19 cell types in the longitudinal scRNA-seq data of $n = 4$ healthy participants. The CVs for each of the $n = 4$ participants were evaluated based on pseudo-bulk intensities in the corresponding 6 independent peripheral blood mononuclear cell (PBMC) samples. The 93 STATIC genes include up to ten top STATIC genes from individual cell types. **b** Circos plots displaying CV of five example STATIC genes identified from each of five major cell types: T cells, B cells, natural killer (NK) cells, monocytes (Mono), and dendritic cells (DCs). **c** Uniform Manifold Approximation and Projection (UMAP) using only the 220 STATIC genes as input features (sUMAP) on the same longitudinal scRNA-seq data. **d–f** sUMAP using the same 220 STATIC genes on three external PBMC datasets ((**d**) CNP0001102[3], (**e**) GSE149689[2], (**f**) GSE164378[16]), where cells are labeled as in the original studies.

**g** Distributions of Pearson correlation coefficient between gene expression (pseudo-bulk intensity) in scRNA-seq data and gene score in scATAC-seq data, one for the 220 STATIC genes (median correlation 0.70), one for the top 500 highly variable genes (HVGs, median correlation 0.40), one for the 10,608 reliable genes (average expression ≥0.1, median correlation 0.21), and one for random gene pairs (95% upper confidence bound at 0.399). The correlations were calculated across 14 cell types in 18 PBMC samples ($n = 252$ data points). **h, i** Venn diagrams showing the overlaps between the 220 STATIC genes and biomarkers distinguishing either healthy controls (Normal) versus participants infected with influenza (FLU, left panel) or Normal versus participants infected with SARS-CoV-2 (COVID19, right panel). The biomarkers were identified from either (**h**) CNP0001102[3] or (**i**) GSE149689[2]. Adobe Illustrator (version 27.1.1; https://www.adobe.com/products/illustrator.html) was used to arrange panels and edit text. Source data are provided as a Source Data file.

type-specific stability patterns and have well-documented roles in the corresponding cell types (Supplementary Data 5c).

We used the 220 STATIC genes as input features and projected PBMCs in our scRNA-seq data onto a two-dimensional Uniform

Manifold Approximation and Projection[11] (UMAP; Fig. 4c), which we refer to as sUMAP from now on. We also generated sUMAPs using the same 220 STATIC genes on three independent scRNA-seq datasets[2,3,16] of PBMCs (Fig. 4d–f) in which cells were labeled as in the original

studies. In all four cases, the 220 STATIC genes separated major cell types and most of their subtypes very well, suggesting that some STATIC genes are potentially good markers for cell types.

Gene scores are routinely computed from scATAC-seq data to infer expression of the corresponding genes and used to label cells in scATAC-seq data based on a scRNA-seq reference[21]. We calculated the Pearson correlation between expression in scRNA-seq data and gene score in scATAC-seq data of the same genes across cell types and samples. Due to data sparseness, incomplete reference assembly, non-coding RNAs, and uncharacterized sequences, Pearson correlation could be calculated only on 10,608 (94.7%) of the 11,191 reliable genes (Fig. 4g). Interestingly, among genes with strong correlations (Supplementary Fig. 10), the correlation was mainly influenced by differences between cell types, which partially justifies the use of gene score for cell labeling on scATAC-seq data. Within individual cell types, the correlation however appeared to be poor across different samples, likely reflecting the complexity of gene regulation. Pearson correlation was obtained on 206 (93.6%) of the 220 STATIC genes with a median value of 0.70. In comparison, Pearson correlation was obtained on 403 (80%) of the top 500 highly variable genes (HVGs), which are widely used in dimension reduction on scRNA-seq data[11], with a significantly lower median value of 0.40 ($p = 1.98 \times 10^{-13}$, Mann–Whiney test; Supplementary Data 5d). We randomly paired unrelated genes, calculated the corresponding correlations between expression and gene score, and found that the obtained distribution had a 95% upper confidence bound at $R_0 = 0.399$ (Methods). Thus, any correlations below $R_0$ were not statistically better than those between random, unrelated gene pairs. A total of 7252 (68%) out of the 10,608 reliable genes and 201 (49.8%) out of the 403 HVGs had a correlation below $R_0$, in comparison with 40 (19%) out of the 206 STATIC genes. To properly label cells in scATAC-seq data based on gene score approach, one should only use genes whose expression versus gene score correlations are above $R_0$. Some STATIC genes might be good candidates for this purpose.

We further investigated how the 220 STATIC genes fared as potential disease biomarkers. Previously, two studies[2,3] applied scRNA-seq to analyze PBMCs of healthy controls (Normal) and of patients infected with either influenza (FLU) or SARS-CoV-2 (COVID19). We reanalyzed the data using methods described in the original studies and identified differential expression genes (DEGs) distinguishing Normal versus FLU or Normal versus COVID19. For simplicity, DEGs from individual cell types were combined when compared with the 220 STATIC genes. Out of the 18,824 genes measured in the first study (CNP0001102)[3], 681 and 632 DEGs were identified for distinguishing Normal versus FLU and Normal versus COVID19, respectively. The corresponding overlap with the STATIC genes was 49 for Normal versus FLU (one-side hypergeometric $p = 4.8 \times 10^{-26}$) and 50 for Normal versus COVID19 (one-side hypergeometric $p = 1.7 \times 10^{-28}$, Fig. 4h). A total of 33,538 genes were measured in the second study (GSE149689)[2]. A total of 126 STATIC genes (one-side hypergeometric $p = 4.8 \times 10^{-74}$) overlapped with the 3040 DEGs for Normal versus FLU while 86 STATIC genes (one-side hypergeometric $p = 2.1 \times 10^{-61}$) overlapped with the 1396 DEGs for Normal versus COVID19 (Fig. 4i). In all cases, the 220 STATIC genes were significantly enriched as DEGs, suggesting their potential for monitoring some disease conditions.

To illustrate that SPECT can handle scRNA-seq data of diverse sample types, we applied it to scRNA-seq data from a mouse brain study (GSE129788)[36]. In the study scRNA-seq data was collected from brain tissues of eight young (2–3 months) and eight old (21–23 months) mice, from which 37,069 cells of high-quality data were labeled to 25 cell types, 14,699 genes were detected, marker genes for each of the 25 cell types were collected, and 1113 DEGs distinguishing young versus old mouse brains were identified from a subset of 15 cell types. The study was not longitudinal per se. We treated data from the eight samples of each age group as repeated measurements for the group, just like repeated measurements at

different timepoints in a longitudinal study. Since SPECT does not utilize the ordering of timepoints, its usage to the data is justified. We collected up to 25 STATIC genes per cell type and obtained 304 unique genes from all 25 cell types (Fig. 5a, Supplementary Data 6a). sUMAP using these 304 STATIC genes was able to separate the cell types as labeled in the original study very well (Fig. 5b). Out of the 304 STATIC genes, 299 genes were identified in the original study as marker genes for the corresponding cell types (Fig. 5c, Supplementary Data 6b). From the 15 cell types having DEGs, we collected 234 STATIC genes that were significantly overlapped with the 1113 young versus old DEGs ($n = 123$, one-side hypergeometric $p = 6.2 \times 10^{-77}$, Fig. 5d). These results further demonstrated that some STATIC genes are good markers for cell types or biological conditions in the mouse brain study.

## Circos plots to reveal stability patterns of protein families

PALMO implements circos plots to display stability patterns from multiple single-cell data modalities together. We displayed the stability pattern of gene expression and gene score of six protein families that are essential for immunity in Fig. 6, including human leukocyte antigens (HLAs, Fig. 6a), interferon regulatory factors (IRFs, Fig. 6b), interleukins (ILs, Fig. 6c), chemokine (C-X-C motif) receptor/ligand (CXCR/L) family (Fig. 6d), Janus kinases (JAKs) and signal transducer and activator of transcription proteins (STATs, Fig. 6e), and tumor necrosis factor receptor superfamily (TNFRSF, Fig. 6f). All these protein families showed diverse stability patterns among members and across cell types, with HLAs and ILs having the most striking contrasts. The rich variety in such stability patterns suggests that different members of same protein superfamilies may play different roles in individual cell types. We noticed that gene expression and gene score generally did not exhibit the same stability patterns despite the rather strong correlations between them (Supplementary Fig. 11). It turns out that strong correlations were mainly driven by difference between cell types rather than difference between samples, likely reflecting the complexity of gene regulation as mentioned before.

## Application of TCA to reveal heterogenous immune responses among COVID-19 patients

We applied TCA to analyze longitudinal scRNA-seq data of four COVID-19 patients, each having data of at least three timepoints, in a previous study[3] and identified significantly up- or down-regulated genes over time (adjusted $p < 0.05$ and slope magnitude >0.1, Fig. 7a–d, Supplementary Data 7a) and the corresponding pathways (Supplementary Data 7b). We observed rather heterogeneous immune responses by these patients during recovery (Fig. 7e), which was not presented in the original study.

Patient COV-3 had barely any significant genes except that IFI27 decreased in DCs, IFI44L decreased in naïve B cells, and IGLC3 decreased in plasma cells, suggesting possible dampening of immune modulation.

The significant genes of patient COV-2 included eighteen upregulated genes in monocytes, four genes each in memory B cells and naïve B cells, and twelve genes split among other six cell types. Gene enrichment analysis on the eighteen upregulated genes in monocytes revealed only one significant pathway: myeloid leukocyte mediated immunity (adjusted $p = 0.044$).

The significant genes of COV-1 included eleven upregulated and six downregulated genes in cycling plasma cells, seven upregulated and sixteen downregulated genes in cycling T cells, six downregulated genes in naïve B cells, and fifteen genes split among other seven cell types. The significant genes in cycling plasma cells were significantly enriched in five pathways including regulation of humoral immune response (adjusted $p = 3.92 \times 10^{-3}$), Fc receptor mediated stimulatory signaling pathway (adjusted $p = 3.92 \times 10^{-3}$), and immunoglobulin production (adjusted $p = 0.011$), indicating a predominant role of humoral immunity in the recovery of the patient.

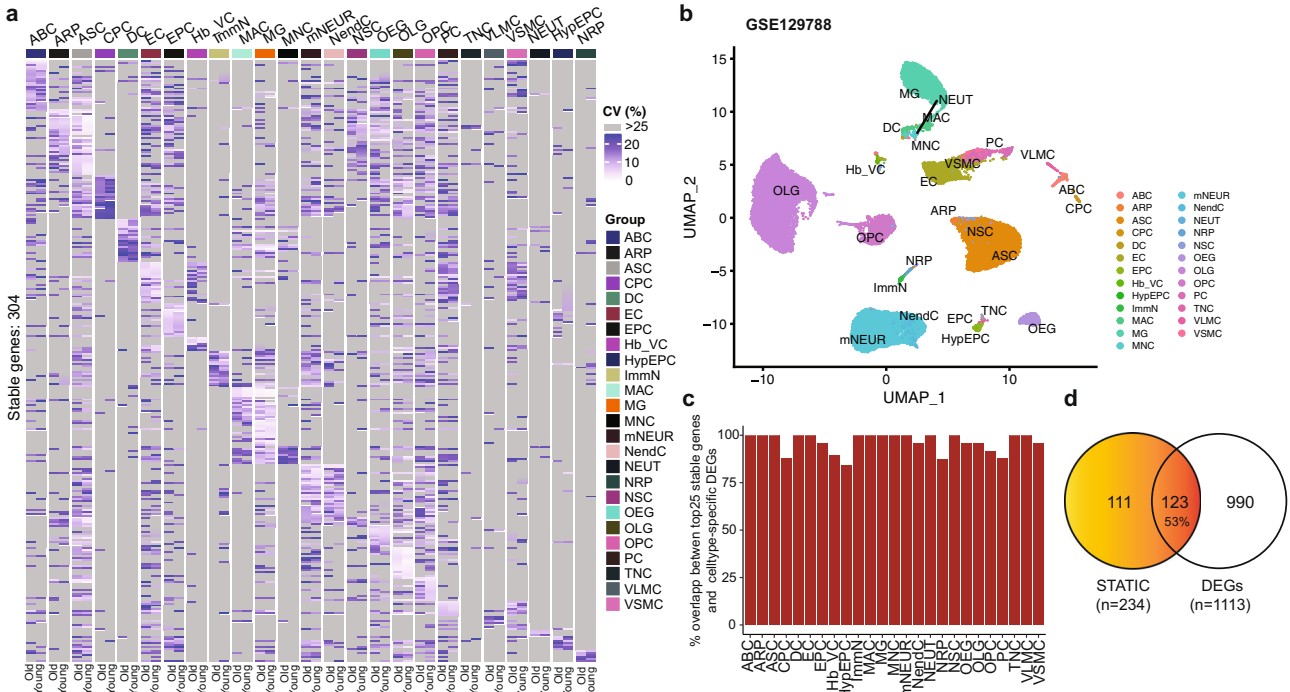

**Fig. 5 | Properties of 304 STATIC genes of mouse brain tissue. a** Heatmap of coefficient of variation (CV) of the 304 stable across time in cell-types (STATIC) genes that were identified from 25 cell types in the scRNA-seq data of a mouse brain study (GSE129788[36]). The CVs were evaluated based on pseudo-bulk intensities in brain tissues from either $n = 8$ young or $n = 8$ old mice. **b** Uniform Manifold Approximation and Projection (UMAP) using only the 304 STATIC genes as input features (sUMAP) on the same scRNA-seq data. Cells are labeled as in the original study. **c** Percentage of top STATIC genes that overlap with cell-type

marker genes identified in the original study. Up to 25 top STATIC genes from each cell type are compared with the corresponding marker genes of the same cell type. **d** Venn diagram showing the overlap between the 234 STATIC genes identified from 15 out of the 25 cell types and biomarkers distinguishing young versus old mice that were identified in the original study from the same 15 cell types. Adobe Illustrator (version 27.1.1; https://www.adobe.com/products/illustrator.html) was used to arrange panels and edit text. Source data are provided as a Source Data file.

Patient COV-5 had significant genes in almost all cell types except for DCs and monocytes, including eight upregulated and eight downregulated genes in memory B cells, six upregulated and six downregulated genes in naïve B cells, one upregulated and ten downregulated genes in activated CD4$^+$ T cells, two upregulated and eight downregulated genes in plasma cells, and 43 genes split among other seven cell types. Seven (58%) of the twelve significant genes in naïve B cells were also significant in memory B cells and in the same direction of change, suggesting common responses by the two cell types. The significant genes in memory B cells were enriched in interferon gamma (adjusted $p = 3.28 \times 10^{-6}$) and alpha (adjusted $p = 4.86 \times 10^{-5}$) response, antigen processing and presentation (adjusted $p = 0.036$), and antigen processing and presentation of peptide or polysaccharide antigen via MHC class II (adjusted $p = 0.044$). The significant genes in naïve B cells were enriched in interferon alpha (adjusted $p = 1.96 \times 10^{-5}$) and gamma (adjusted $p = 1.96 \times 10^{-5}$) response. The significant genes in plasma cells were enriched in innate and humoral immune responses ($p = 3.46 \times 10^{-4}$ and $p = 5.79 \times 10^{-4}$, respectively) although both with an adjusted $p = 0.084$. These results aligned to the patient's disease severity and advanced age.

For comparison, we also used Seurat to analyze patient COV-5 data of activated CD4$^+$ T cells. To satisfy Seurat's requirement of selecting two contrast groups, we did the analysis in two iterations, i.e., day 1 (D1) versus D7 + D13 and D1 + D7 versus D13. We obtained 942 and 1018 DEGs (adjusted $p < 0.05$), respectively, with an overlap of 813 DEGs (Supplementary Fig. 12a). TCA identified 921 significantly up- or down-regulated genes (adjusted $p < 0.05$), only 21 of which overlapped with both Seurat results. The genes obtained from TCA or Seurat were quite different. We collected top ten up- and top ten down-regulated genes from all three approaches and plotted the corresponding gene

expression in heatmaps (Supplementary Fig. 12b–d). TCA results showed better dynamic changes over time than Seurat results in our opinion.

## Discussion

The five modules in PALMO analyze longitudinal omics data from multiple perspectives as continuous data. VDA provides a global view on the sources of variance within the whole dataset. TCA studies the time series of individual participants. CVP and SPECT first examine data of individual participants separately and then summarize the observations across different participants. All these four methods focus on individual features. ODA is the only method to provide a sample-level analysis. Which module(s) to use on a specific dataset depends on the research question of interest. Additional methods need to be developed for research interest not covered here.

We observed that a small set of STATIC genes, 220 for PBMC and 304 for mouse brain tissues, distinguished cell types well and captured some biological differences. The PBMC STATIC genes showed better correlation between gene expression in scRNA-seq data and gene score in scATAC-seq data than HVGs. It would be interesting to see whether these observations can be extended to scRNA-seq data of other sample types. We selected up to 25 STATIC genes per cell type in our analysis. It is possible that a better set of genes can be selected with a more sophisticated selection procedure.

Plasma proteins are often targeted as disease biomarkers, thus understanding their temporal stability is of particular interest. Conceptually, highly variable proteins are poor biomarker candidates since their values likely have very high sampling variations. The rather moderate CV values of the most variable proteins in our study suggest sampling variations are not a big concern on these proteins. The small

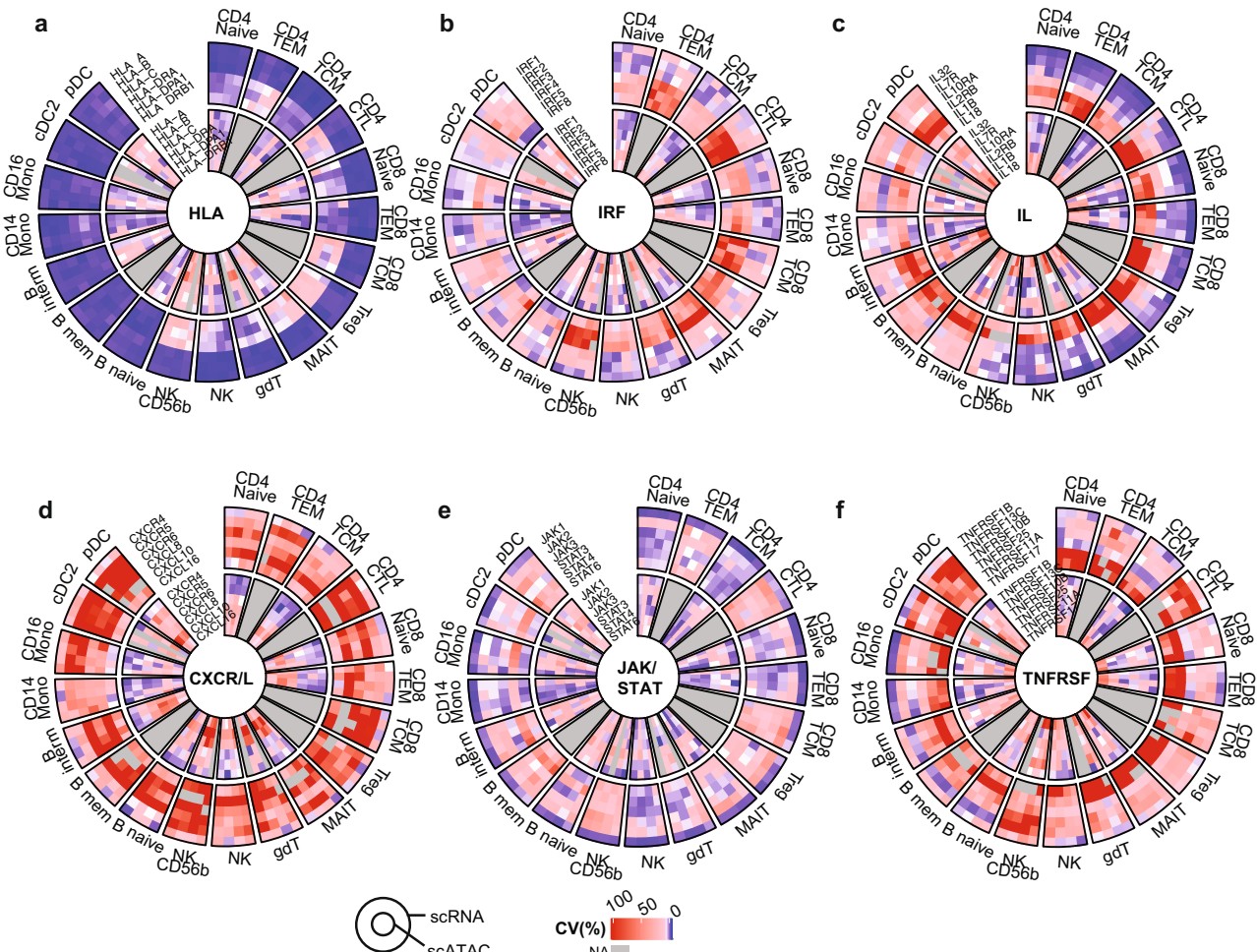

**Fig. 6 | Circos plots showing stability patterns of five protein families. a** Circos plot displaying stability patterns of gene expression (outer circles) and gene score (inner circles) of human leukocyte antigen (HLA) protein family (member: HLA-A, HLA-B, HLA-C, HLA-DRA, HLA-DPA1, and HLA-DRB1). Samples with missing data or cell types with low cell counts are shown in grey. **b**–**f** Same as (**a**) but for (**b**) interferon regulatory factors (IRFs; member: IRF1, IRF2, IRF3, IRF4, IRF5, and IRF8), (**c**) interleukins (ILs; member: IL32, IL7R, IL10RA, IL2RB, IL1B and IL18), (**d**) chemokine (C-X-C motif) receptor/ligand (CXCR/L) protein family (member: CXCR4, CXCR5, CXCR6, CXCL8, CXCL10, and CXCL16), (**e**) Janus kinase (JAK) and signal transducer and activator of transcription (STAT) protein family (member: JAK1,

JAK2, JAK3, STAT3, STAT4, and STAT6), and (**f**) tumor necrosis factor receptor superfamily (TNFRSF; member: TNFRSF1B, TNFRSF13C, TNFRSF10B, TNFRSF25, TNFRSF11A, and TNFRSF17). The CV of gene expression for each of $n = 4$ participants was calculated from pseudo-bulk intensities in the corresponding 6 independent peripheral blood mononuclear cell (PBMC) samples. The CV of gene score for each participant was based on either 6 (for $n = 2$ participants) or 3 (for other $n = 2$ participants) PBMC samples. Adobe Illustrator (version 27.1.1; https://www.adobe.com/products/illustrator.html) was used to arrange panels and edit text. Source data are provided as a Source Data file.

CV values of the most stable proteins, on the other hand, indicate they do not change much under normal, healthy conditions. So, if they ever change under some disease conditions, they should be closely explored as potential biomarkers.

We condensed single-cell data into pseudo-bulk data in VDA, SPECT and ODA. Recent literature[14,17,18] revealed that many single-cell methods fail to properly account for variations in cross-sectional scRNA-seq data and generate many false DEGs as a result. In comparison, pseudo-bulk approaches mostly generate reliable results although they may be underpowered. Longitudinal single-cell omics data is even more complicated than cross-sectional scRNA-seq data and may require new statistical methods to properly handle its many types of variations. Furthermore, memory and CPU requirements for using GLMMs to analyze longitudinal single-cell omics data at single-cell level may be challenging even for cloud-based computing. We adopted the pseudo-bulk approach in VDA, SPECT and ODA as a practical compromise. In TCA we bypassed some of the complications by analyzing data of individual cell types and of individual participants separately.

The lack of a well-accepted software package for longitudinal omics data makes it difficult to benchmark PALMO performance. We compared PALMO with variancePartition[19], tcR[20], and Seurat[16], which is summarized in Supplementary Fig. 1c. VDA can handle missing data but variancePartition cannot, which is an advantage of VDA since missing values in longitudinal omics data are almost inevitable. The two tools generated almost identical results on two tested datasets after removing missing values. PALMO was not developed specifically for TCR data. When we applied VDA to the TCR data of SSc donors, we obtained results that are potentially interesting but not reported in the original study using tcR. We believe PALMO complements TCR specific tools (such as tcR) on TCR data. Seurat requires users to select two contrast groups in DEG analysis and thus is not appropriate for analyzing longitudinal data of more than two timepoints. Nevertheless, when we applied both TCA and Seurat to the longitudinal scRNA-seq data of activated CD4$^+$ T cells of a COVID-19 patient, the two methods generated rather different results on up- or down-regulated genes. Heatmaps of the corresponding top genes revealed that TCA results showed better dynamic changes over time than Seurat results.

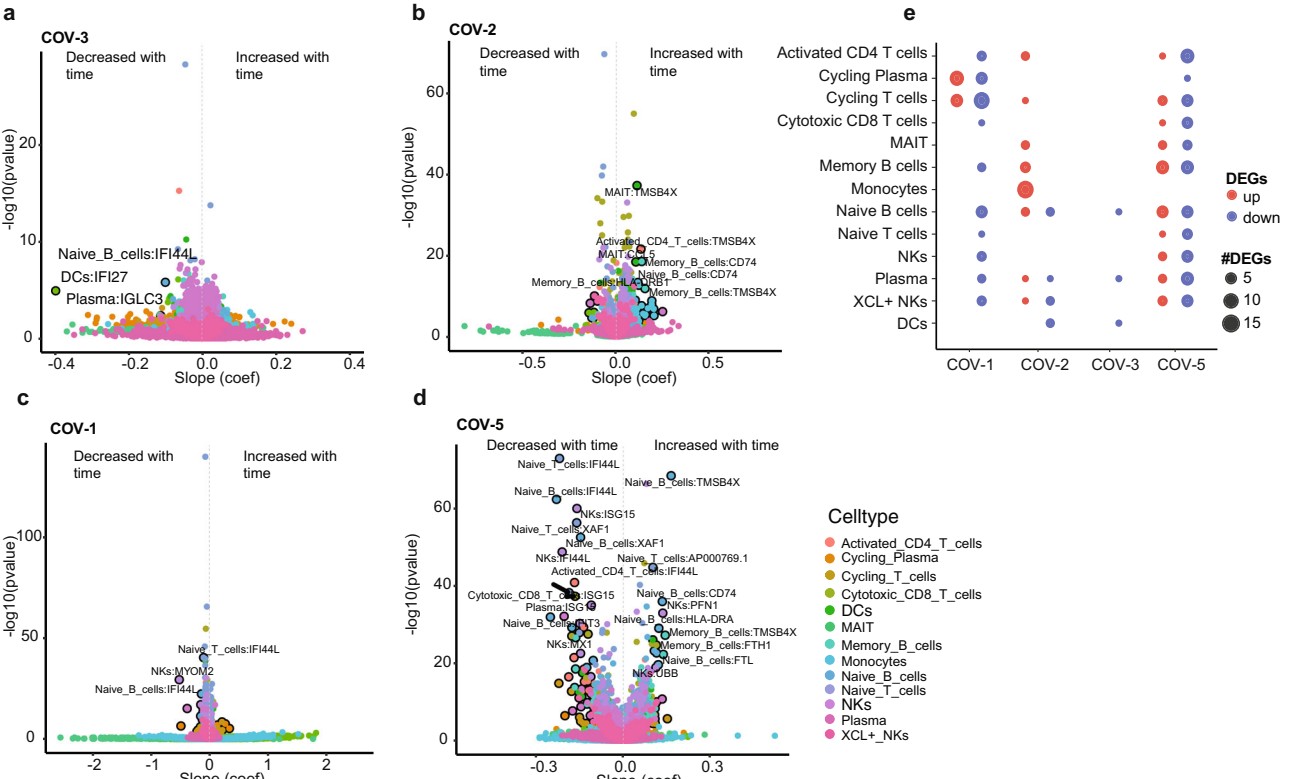

**Fig. 7 | Heterogeneous immune responses by COVID19 patients during recovery. a** Volcano plot showing temporal expression changes of individual genes in different cell types during the recovery of patient COV-3 (female, 41 years old, mild symptoms, data on day D1/D4/D16), based on longitudinal scRNA-seq data in CNP0001102[3]. The x-axis shows the slope (coefficient) of gene expression change as a linear function of time. The y-axis shows the corresponding adjusted p value of the slope. **b–d** Same as (**a**) but for patients (**b**) COV-2 (male, 45 years old, mild symptoms, data on D1/D4/D7/D10/D16), (**c**) COV-1 (male, 15 years old, mild symptoms, data on D1/D4/D16), and (**d**) COV−5 (female, 85 years old, severe symptoms, data on D1/D7/D13). **a–d** Each plot contains results on up to 18,824 genes in 13 cell types (up to 244,712 data points). **e** Counts of significantly upregulated (adjusted $p<0.05$ and $slope>0.1$, red) and significantly downregulated (adjusted $p<0.05$ and $slope<−0.1$, blue) genes during the recovery of the four COVID-19 patients in individual cell types. **a–d** The $p$-value for slope was calculated based on two-sided likelihood-ratio test and adjusted by Benjamini and Hochberg procedure for testing many genes. Adobe Illustrator (version 27.1.1; https://www.adobe.com/products/illustrator.html) was used to arrange panels and edit text. Source data are provided as a Source Data file.

PALMO has been published as an R package in CRAN with a detailed reference manual and vignettes to demonstrate its usage. It can be easily installed and executed in R or RStudio. As we demonstrated, it can be used to analyze longitudinal bulk and single-cell omics data generated on diverse technical platforms and/or of diverse sample types, including but not limited to: clinical lab test results, cell type composition, gene expression, protein abundance, bulk or single-cell omics data, TCR sequencing data, etc. We believe it can facilitate the analysis of some longitudinal omics data. In addition, our longitudinal multi-omics dataset of five data modalities on the same samples can also be a valuable resource for immune health study and software development.

## Methods
### Healthy donors
Blood samples were obtained from Bloodworks Northwest (Seattle, WA) through protocols approved by the Bloodworks Northwest institutional review board and complying with all relevant ethical regulations. We enrolled $n = 6$ clinically healthy participants (no diagnosis of active or chronic disease) with age between 25 to 38 years with equal self-report sex ratio. Viable peripheral blood mononuclear cells (PBMCs) and plasma samples were collected from each participant over $t = 10$ weeks. Complete blood count (CBC) was measured to evaluate overall health of all donors over all timepoints ($n = 6$, t = 10). Minimal biometric data were collected on these participants which were handled following the Health Insurance Portability and

Accountability Act (HIPAA) guidelines. Informed consent to participate in the study and to publish data from the research was obtained from all participants.

### Sample handling
A volume of 30 mL of blood was drawn into BD NaHeparin vacutainer tubes (for PBMC; BD #367874) or K2-EDTA vacutainer tubes (for plasma; BD #367863). Upon arrival at the processing lab all NaHeparin tubes for each donor were pooled into a sterile plastic receptacle to establish one common pool and stored at room temperature until processing (4 h or less from draw). For PBMC isolation, at each time point the pool of blood was gently swirled until fully mixed, about 30 times, and a volume of blood was removed and combined with an equivalent volume of room temperature PBS (ThermoFisher #14190235). PBMC were isolated using one or more Leucosep tubes (Greiner Bio-One #227290) loaded with 15 mL of Ficoll Premium (GE Healthcare #17-5442-03) to which a 3 mL cushion of PBS had been slowly added on top of the Leucosep barrier. The 24−30 mL diluted whole blood was slowly added to the tube and spun at 1000xg for 10 min at 20 °C with no brake. PBMC were recovered from the Leucosep tube by quickly pouring all volume above the barrier into a sterile 50 mL conical tube; 15 mL cold PBS + 0.2% BSA (Sigma #A9576; "PBS + BSA") was added, and the cells were pelleted at 400xg for 5–10 min at 4–10 °C. The supernatant was quickly decanted, the pellet dispersed by flicking the tube, and the cells washed with 25–50 mL cold PBS + BSA. Cell pellets were combined, if applicable,

the cells were pelleted as before, supernatant quickly decanted, and residual volume was carefully aspirated. The PBMC were resuspended in 1 mL cold PBS + BSA per 15 mL whole blood processed and counted with a Cellometer Spectrum (Nexcelom) using Acridine Orange/Propidium Iodide solution. PBMC were cryopreserved 90% FBS (ThermoFisher #10438026)/10% DMSO (Fisher Scientific #D12345) at $1-5 \times 10^6$ cells/mL by slow freezing in a Coolcell LX (VWR #75779-720) overnight in a −80 °C freezer followed by transfer to liquid nitrogen.

For plasma isolation, the K2-EDTA source tube was gently inverted 10 times, and the appropriate volume of whole blood was extracted using an 18-gauge needle and syringe and transferred to a similar plastic tube with no additives (Greiner Bio-One #456085). The blood was centrifuged at 2000xg for 15 min at 20 °C with a brake of 1, and 80%–90% of the plasma supernatant were removed by careful pipetting for immediate freezing at −80 °C. Plasma was assayed after the first freeze/thaw. Thawed PBMC of four donors over six time-points ($n = 4$, $t = 6$) were assayed by flow cytometry, scRNA-seq and scATAC-seq in two batches (donors PTID5 and PTID6, donors PTID2 and PTID4) by a team of operators. Plasma of all donors over all timepoints ($n = 6$, $t = 10$) was isolated and cryopreserved by a team of operators[37].

## Flow cytometry

PBMC were removed from liquid nitrogen storage and immediately thawed in a 37 °C water bath. Cells were diluted dropwise into 37 °C AIM V media (Thermo Fisher Scientific #12055091) up to a final volume of 10 mL. A single wash was performed in 10 mL of PBS + BSA, pelleting cells at 400xg for 5–10 min at 4–10 °C. PBMC were resuspended 2 mL in PBS + BSA and counted using a Cellometer Spectrum. $1-2 \times 10^6$ cells were incubated with Human TruStain FcX (BioLegend #422302) and Fixable Viability Stain 510 (BD #564406) prior to staining with a 25-color cell surface panel on ice for 25 min. Cells were washed and fixed with 4% paraformaldehyde (Electron Microscopy Sciences #15713) prior to acquisition on a BD Symphony cytometer. Raw data were compensated and curated to remove unrepresentative events due to instrument fluidics variability (time gating), doublets (by FSC-H and FSC-W), and cells exhibiting membrane permeability (live/dead gating) prior to quantification using BD FlowJo software v10.6.1.

## Proteomics

Plasma samples were submitted to Olink (Uppsala, Sweden) for assay using the Olink Proximity Extension assay, run on the Fluidigm Biomark system. Patient samples were distributed evenly across two plates, and all time points per patient were run on the same plate, with randomized well locations. Samples were assayed using the Olink Discovery Assay which encompasses a total of 1156 proteins across 13 panels (Cardiometabolic [V.3603], Cardiovascular II [V.5006], Cardiovascular III [V.6113], Cell Regulation [V.3701], Development [V.3512], Immune Response [V.3202], Inflammation [V.3021], Metabolism [V.3402], Neuro Exploratory [V.3901], Neurology [V.8012], Oncology II [V.7004], Oncology III [V.4001], Organ Damage [V.3311]). Quality assessment, limit of detection, and normalization were performed by Olink using the plate bridging control, two positive controls, and three background controls.

## Single-cell RNA-seq

**Sample preparation, hashing, and pooling.** Single-cell RNA-seq libraries were generated on PBMC prepared as above using the 10x Genomics Chromium 3′ Single Cell Gene Expression assay (#1000121) and Chromium Controller Instrument according to the manufacturer's published protocol with modifications for cell hashing[38]. To block off-target antibody binding, Blocking Solution (5 μL of Human Trustain FcX (BioLegend #422302), and 13.7 μL of a 10% Bovine Serum Albumin (BSA)) was added to 500,000 cells suspended in 50 μL Dulbecco's

Phosphate Buffered Saline (DPBS; Corning Life Sciences #21-031-CM) and incubated for 10 min on ice. To stain samples, 0.5 μg (1 μL) of a TotalSeq™-A anti-human Hashtag Antibody was suspended in 31.3 μL DPBS/2% BSA, then added to each sample. For each batch of samples, 100,000 cells from 12 hashed samples with a distinct Hashtag Antibody were pooled into the hashed pool. Roughly 20,000 cells from a Leukopak healthy control were also labeled with a distinct TotalSeq™-A Hashtag Antibody and were spiked into each pool to serve as a batch control.

**Droplet encapsulation and reverse transcription.** From each pool, 64,000 cells were loaded into each well of a Chromium Single Cell Chip G (10x Genomics #1000073) (8 wells per chip), targeting a recovery of 20,000 singlets from each well. Gel Beads-in-emulsion (GEMs) were then generated using the 10x Chromium Controller. The resulting GEM generation products were then transferred to semi-skirted 96-well plates and reverse transcribed on a C1000 Touch Thermal Cycler (Bio-Rad) programmed at 53 °C for 45 min, 85 °C for 5 min, and a hold at 4 °C. Following reverse transcription, GEMs were broken, and the pooled single-stranded cDNA and Hashtag Oligo fractions were recovered using Silane magnetic beads (Dynabeads MyOne SILANE #37002D).

**Library generation and separation.** Barcoded, full-length cDNA including the Hashtag Oligos (HTOs) from the TotalSeq™-A Hashtag Antibodies were then amplified with a C1000 Touch Thermal Cycler programmed at 98 °C for 3 min, 11 cycles of (98 °C for 15 s, 63 °C for 20 s, 72 °C for 1 min), 72 °C for 1 min, and a hold at 4 °C. Amplified cDNA was purified and separated from amplified HTOs using a 0.6x size selection via SPRIselect magnetic bead (Beckman Coulter #22667) and a 1:10 dilution of the resulting cDNA was run on a Fragment Analyzer (Agilent Technologies #5067-4626) to assess cDNA quality and yield. HTO libraries were purified further with SPRIselect magnetic bead (Beckman Coulter #22667) and amplified and indexed with a custom HTO i7 index on a C1000 Touch Thermal Cycler programmed at 95 °C for 3 min, 10 cycles of (95 °C for 20 s, 64 °C for 30 s, 72 °C for 20 s), 72 °C for 1 min, and a hold at 4 °C. The resulting HTO libraries were purified with SPRIselect magnetic bead (Beckman Coulter #22667) post-amplification and a 1:10 dilution of the resulting HTO libraries were run on a Fragment Analyzer (Agilent Technologies #5067-4626) to assess HTO quality and yield. A quarter of the cDNA sample (10 ul) was used as input for library preparation. Amplified cDNA was fragmented, end-repaired, and A-tailed is a single incubation protocol on a C1000 Touch Thermal Cycler programmed at 4 °C start, 32 °C for min, 65 °C for 30 min, and a 4 °C hold. Fragmented and A-tailed cDNA was purified by performing a dual-sided size-selection using SPRIselect magnetic beads (Beckman Coulter #22667). A partial TruSeq Read 2 primer sequence was ligated to the fragmented and A-tailed end of cDNA molecules via an incubation of 20 °C for 15 min on a C1000 Touch Thermal Cycler. The ligation reaction was then cleaned using SPRIselect magnetic beads (Beckman Coulter #22667). PCR was then performed to amplify the library and add the P5 and indexed P7 ends (10x Genomics #1000084) on a C1000 Touch Thermal Cycler programmed at 98 °C for 45 sec, 13 cycles of (98 °C for 20 sec, 54 °C for 30 sec, 72 °C for 20 sec), 72 °C for 1 min, and a hold at 4 °C. PCR products were purified by performing a dual-sided size-selection using SPRIselect magnetic beads (Beckman Coulter #22667) to produce final, sequencing-ready libraries.

**Quantification and sequencing.** Final libraries were quantified using Picogreen and their quality was assessed via capillary electrophoresis using the Agilent Fragment Analyzer HS DNA fragment kit and/or Agilent Bioanalyzer High Sensitivity chips. Libraries were sequenced on the Illumina NovaSeq platform using S4 flow cells. Read lengths were 28 bp read1, 8 bp i7 index read, 91 bp read2.

**scRNA-seq data pre-processing.** scRNA-seq data of four donors were generated in two batches, each containing data of two donors. Each batch of data was pre-processed separately[37]. Briefly, binary base call (BCL) files were demultiplexed using the mkfastq function in the 10x Cell Ranger software (version 3.1.0), producing fastq files. Fastq files were then checked for quality (FastQC version 0.11.3) and run through the 10x Cell Ranger alignment function (cell ranger count) against the human reference annotation (Ensembl GRCh38). Mapping was performed using default parameters. Upon completion, Cell Ranger produced an output directory per file that contains the following: bam file (binary alignment file), HDF5 file (Hierarchical Data Format) with all reads, HDF file containing just the filtered reads, summary report (html and csv), and cloupe.cloupe (a file for the 10x Loupe visual browser).

**scRNA-seq data analysis.** Individual HDF5 files (filtered) were loaded into the R statistical programming language (version 3.6.0) using Bioconductor (version 3.1.0) and the Seurat package (version 3.1.5)[37]. For simplicity, sample names were captured as a list in R and iteratively processed within a loop (refer to https://satijalab.org/seurat/ for more information). Within the loop, samples were normalized with the NormalizeData function followed by the FindVariableFeatures function with parameters: vst selection method and 2000 features. Label transfer was performed using previously published procedures[39] and with the Seurat reference dataset. Labeling included the FindTransferAnchors and TransferData functions performed in the Seurat package.

We merged the two batches of data using the Seurat *merge* function. We calculated read depth, mitochondrial percentage, and number of UMIs per sample. Cells were filtered with nFeature_RNA > 200 and percent.mt <10. The merged data structure was normalized (using NormalizeData and FindVariableFeatures functions) and then saved as an RDS for further analysis. The top 3000 variable genes were used for PCA and UMAP based dimension-reduction maps using 30 principal components (PCs). We checked for possible batch effects using the bridging controls but did not observe any obvious batch effects.

Cell labels obtained from the original batches were kept. Doublets were removed from further analysis. In total we retrieved high quality data of 472,464 cells from 4 donors and labeled them to 31 cell types from Seurat level 2 labelling. The cell type frequencies in each sample were calculated and compared with flow-based cell frequencies. Nineteen cell types (CD4_Naive, CD4_TEM, CD4_TCM, CD4_CTL, CD8_Naive, CD8_TEM, CD8_TCM, Treg, MAIT, gdT, NK, NK_CD56bright, B_naive, B_memory, B_intermediate, CD14_Mono, CD16_Mono, cDC2, pDC) were selected for further analysis after filtering out cell types with a low frequency (<0.5%).

## Single-cell ATAC-seq

**Sample preparation.** Permeabilized-cell scATAC-seq was performed. A 5% w/v digitonin stock was prepared by diluting powdered digitonin (MP Biomedicals, 0215948082) in DMSO (Fisher Scientific, D12345), which was stored in 20 μL aliquots at −20 °C until use. To permeabilize, $1 \times 10^6$ cells were added to a 1.5 mL low binding tube (Eppendorf, 022431021) and centrifuged (400×g for 5 min at 4 °C) using a swinging bucket rotor (Beckman Coulter Avanti J-15RIVD with JS4.750 swinging bucket, B99516). Cells were resuspended in 100 μL cold isotonic Permeabilization Buffer (20 mM Tris-HCl pH 7.4, 150 mM NaCl, 3 mM MgCl2, 0.01% digitonin) by pipette-mixing 10 times, then incubated on ice for 5 min, after which they were diluted with 1 mL of isotonic Wash Buffer (20 mM Tris-HCl pH 7.4, 150 mM NaCl, 3 mM MgCl2) by pipette-mixing five times. Cells were centrifuged (400 × g for 5 min at 4 °C) using a swinging bucket rotor, and the supernatant was slowly removed using a vacuum aspirator pipette. Cells were resuspended in a chilled TD1 buffer (Illumina, 15027866) by pipette-mixing to a target concentration of 2300–10,000 cells per μL. Cells were filtered through 35 μm Falcon Cell Strainers (Corning, 352235) before counting on a Cellometer Spectrum Cell Counter (Nexcelom) using ViaStain acridine orange/propidium iodide solution (Nexcelom, C52-0106-5).

**Tagmentation and fragment capture.** scATAC-seq libraries were prepared according to the Chromium Single Cell ATAC v1.1 Reagent Kits User Guide (CG000209 Rev B) with several modifications. 19,000 cells were loaded into each tagmentation reaction. Permeabilized cells were brought up to a volume of 12 μl in TD1 buffer (Illumina, 15027866) and mixed with 3 μl of Illumina TDE1 Tn5 transposase (Illumina, 15027916). Transposition was performed by incubating the prepared reactions on a C1000 Touch thermal cycler with 96–Deep Well Reaction Module (Bio-Rad, 1851197) at 37 °C for 60 min, followed by a brief hold at 4 °C. A Chromium NextGEM Chip H (10x Genomics, 2000180) was placed in a Chromium Next GEM Secondary Holder (10x Genomics, 3000332) and 50% Glycerol (Teknova, G1798) was dispensed into all unused wells. A master mix composed of Barcoding Reagent B (10x Genomics, 2000194), Reducing Agent B (10x Genomics, 2000087), and Barcoding Enzyme (10x Genomics, 2000125) was then added to each sample well, pipette-mixed, and loaded into row 1 of the chip. Chromium Single Cell ATAC Gel Beads v1.1 (10x Genomics, 2000210) were vortexed for 30 s and loaded into row 2 of the chip, along with Partitioning Oil (10x Genomics, 2000190) in row 3. A 10x Gasket (10x Genomics, 370017) was placed over the chip and attached to the Secondary Holder. The chip was loaded into a Chromium Single Cell Controller instrument (10x Genomics, 120270) for GEM generation. At the completion of the run, GEMs were collected, and linear amplification was performed on a C1000 Touch thermal cycler with 96–Deep Well Reaction Module: 72 °C for 5 min, 98 °C for 30 sec, 12 cycles of: 98 °C for 10 sec, 59 °C for 30 sec and 72 °C for 1 min.

**Sequencing library preparation.** GEMs were separated into a biphasic mixture through addition of Recovery Agent (10x Genomics, 220016), the aqueous phase was retained and removed of barcoding reagents using Dynabead MyOne SILANE (10x Genomics, 2000048) and SPRIselect reagent (Beckman Coulter, B23318) bead clean-ups. Sequencing libraries were constructed by amplifying the barcoded ATAC fragments in a sample indexing PCR consisting of SI-PCR Primer B (10x Genomics, 2000128), Amp Mix (10x Genomics, 2000047) and Chromium i7 Sample Index Plate N, Set A (10x Genomics, 3000262) as described in the 10x scATAC User Guide. Amplification was performed in a C1000 Touch thermal cycler with 96–Deep Well Reaction Module: 98 °C for 45 sec, for 11 cycles of: 98 °C for 20 sec, 67 °C for 30 sec, 72 °C for 20 sec, with a final extension of 72 °C for 1 min. Final libraries were prepared using a dual-sided SPRIselect size-selection cleanup. SPRIselect beads were mixed with completed PCR reactions at a ratio of 0.4x bead:sample and incubated at room temperature to bind large DNA fragments. Reactions were incubated on a magnet, the supernatant was transferred and mixed with additional SPRIselect reagent to a final ratio of 1.2x bead:sample (ratio includes first SPRI addition) and incubated at room temperature to bind ATAC fragments. Reactions were incubated on a magnet, the supernatant containing unbound PCR primers and reagents was discarded, and DNA bound SPRI beads were washed twice with 80% v/v ethanol. SPRI beads were resuspended in Buffer EB (Qiagen, 1014609), incubated on a magnet, and the supernatant was transferred resulting in final, sequencing-ready libraries.

**Quantification and sequencing.** Final libraries were quantified using a Quant-iT PicoGreen dsDNA Assay Kit (Thermo Fisher Scientific, P7589) on a SpectraMax iD3 (Molecular Devices). Library quality and average fragment size was assessed using a Bioanalyzer (Agilent, G2939A) High Sensitivity DNA chip (Agilent, 5067-4626). Libraries were sequenced on the Illumina NovaSeq platform with the following read lengths: 51nt read 1, 8nt i7 index, 16nt i5 index, 51nt read 2.

**scATAC data pre-processing.** scATAC-seq data were available for donor PTID2 and PTID4 at week 2–7 (6 timepoints) and for PTID5 and PTID6 at week 2, 4, and 7. scATAC-seq libraries were processed. In brief, cellranger-atac mkfastq (10x Genomics v1.1.0) was used to demultiplex BCL files to FASTQ. FASTQ files were aligned to the human genome (10x Genomics refdata-cellranger-atac-GRCh38-1.1.0) using cellranger-atac count (10x Genomics v1.1.0) with default settings. scATAC fragments were submitted to the ArchR package to create the ArchR object[21]. Per-cell quality control (QC) was performed using methods as mentioned in ArchR. The QC analysis showed FRiP score (the fraction of reads that fall into a peak) >0.25. The TSS enrichment and log10(nFrags) data showed comparable range across all samples. Doublets were removed using filterDoublets() function. In total we observed 294,623 peaks in 135,566 cells.

**scATAC-seq data analysis.** Using plotEmbedding function in ArchR, embedded IterativeLSI was used to perform UMAP based dimension reduction. Unconstrained integration was used to align scATAC-seq gene score matrix in ArchR object with the corresponding scRNA-seq gene expression matrix, from which cells were labeled to 28 cell types along with labeling scores to measure the quality of the cell-label transfer. We filtered out low quality cells (labeling score <0.5), removed cell types having less than 50 remaining cells, and kept 14 (B_intermediate, B_naive, CD14_Mono, CD16_Mono, CD4_Naive, CD4_TCM, CD8_Naive, CD8_TEM, cDC2, gdT, MAIT, NK, NK_CD56bright, and pDC) out of the 28 cell types for downstream analysis. The gene score matrix was retrieved using the getGroupSE() function in ArchR[21] and used for downstream analysis by PALMO.

## Reagents and resources
Critical reagents and resources used in our experiments are listed in Supplementary Data 8.

## PALMO
**Overview.** The current version of PALMO contains five analytical modules to analyze longitudinal omics data from multiple perspectives. It treats longitudinal omics data as continuous variables. PALMO has been published as an R package in CRAN with a detailed reference manual and vignettes to demonstrate its usage (https://cran.r-project.org/web/packages/PALMO/index.html). It can be easily installed and executed in R or RStudio.

**PALMO S4 object.** PALMO is a R based package that uses the setClass function to create an S4 object oriented system. The S4 object consists of a list of data structures with different types of elements such as strings, numbers, vectors, embedded lists, etc. It stores input expression data, input metadata, and output results into separate data structures for easy retrieval and interpretation. More details can be found in Section 3.9 of PALMO vignettes (https://raw.githubusercontent.com/aifimmunology/PALMO/main/Vignette-PALMO.pdf).

Function *createPALMOobject()* takes two inputs (*anndata* and *data*) to create an PALMO S4 object: *anndata* is a data frame containing sample annotations. For longitudinal bulk data, *data* is a data frame with features (such as genes or proteins) as rows, samples as columns, and expression values as elements. For longitudinal single-cell omics data, *data* is a Seurat object. For single-cell omics data without a Seurat object, function *createPALMOfromsinglecellmatrix()* first creates a Seurat object from an expression matrix or data frame and then creates a PALMO S4 object. Function *annotateMetadata()* assigns columns in the original sample annotation data to designated variables (*sample_column*, *donor_column*, and *time_column*) of the PALMO object for longitudinal analysis. Function *mergePALMOdata()* cleans up the PLAMO object by filtering out data missing essential information on *sample_column*, *donor_column*, or *time_column*. Function *checkReplicates()* first checks whether there are replicated samples

at the same time points and of the same participants and, if yes, takes the median values among replicated samples. Function *avgExpCalc()* carries out pseudo-bulking on single-cell omics data. Function *naFilter()* filters out data whose fraction of NAs is above *na_cutoff* (default: 0.4).

**Variance decomposition analysis (VDA).** For variance decomposition, we want to evaluate contributions from factors of interest $\{F_i\}$ to the total variance of analyte Y with or without the influence of fixed effects $\{X_j\}$. Some $\{F_i\}$ and $\{X_j\}$ may be the same variables. We treat $\{F_i\}$ as random effects in a linear mixed model, that is, with fixed effects,

$$Y \sim X_1 + X_2 + \ldots + X_m + (1|F_1) + (1|F_2) + \ldots + (1|F_n). \quad (1)$$

Or, without fixed effects,

$$Y \sim (1|F_1) + (1|F_2) + \ldots + (1|F_n). \quad (2)$$

Using lme4[40], one can obtain the corresponding variance $\sigma_i^2$, including the residual variance $\sigma_R^2$. Then the total variance of Y is given by

$$\sigma_{total}^2 = \sigma_1^2 + \sigma_2^2 + \ldots + \sigma_n^2 + \sigma_R^2. \quad (3)$$

The proportion of variance explained by factor $F_i$ is then $\sigma_i^2/\sigma_{total}^2$. Similar approach was used in variancePartition[19] where the percentage of variance explained was interpreted as the intra-class correlation (ICC). VDA can be performed with the function *lmeVariance()*. VDA results can be displayed with functions *variancefeaturePlot()* and *gene_featureplot()*.

**Coefficient of variation (CV) profiling (CVP).** CVP is designed for bulk longitudinal data and contains two functions: (1) Function *cvCalcBulkProfile()* calculates CV of all features and generates the corresponding CV profile. (2) Function *cvCalcBulk()* identifies consistently stable and variable features, which has two important parameters: Parameter *cvThreshold* (default: 5%) specifies the CV cutoff for distinguishing stable (CV < *cvThreshold*) or variable (CV > *cvThreshold*) features. Parameter *donorThreshold* (default: the total number of donors) defines the minimum number of donors on which a feature needs to be stable or variable to be considered as consistently stable or variable. One may choose *cvThreshold* as the mode of the corresponding CV distribution.

**Stability pattern evaluation across cell types (SPECT).** SPECT is the CVP counterpart for single-cell data and contains the following functions: (1) Function *cvCalcSCProfile()* calculates the CVs of all features in individual cell types and of individual donors and generates the corresponding CV profile. (2) Function *cvSCsampleprofile()* calculates the CVs of all features of individual donors regardless of difference in cell types and generates the corresponding CV profile. (3) Function *cvCalcSC()* determines whether individual features are stable (CV < *cvThreshold*) or variable (CV > *cvThreshold*) in individual cell types and of individual donors. One may choose *cvThreshold* as the mode of the corresponding CV distribution or a convenient value based on the CVs of housekeeping genes. (4) Function *VarFeatures()* first counts how many times individual features are variable in cell type-donor combinations and then classifies variable features as follows: Features whose counts are above parameter *groupThreshold* are classified as super variable (SUV). Features whose counts are below *groupThreshold* but which are consistently variable across all donors in at least one cell type are classified as variable across time in cell-types (VATIC). The default *groupThreshold* value is set to $N_{donor}*N_{celltype}/2$ where $N_{donor}$ is the number of donors and $N_{celltype}$ is the number of cell types. (5) Function *StableFeatures()* is similar to *VarFeatures()* but classifies stable

features as super stable (SUS) or stable across time in cell-types (STATIC). (6) Function *dimUMAPPlot()* generates a UMAP plot using a set of selected genes as input.

**Outlier detection analysis (ODA).** ODA applies both graphic and statistical methods to examine the temporal behavior of longitudinal data. Function *sample_correlation()* calculates intra- and inter-donor correlations (across analytes) and displays the results in a heatmap. Timepoints showing obvious weaker correlations with other timepoints are potential outliers. To detect abnormal timepoints, function *outlierDetect()* first calculates the mean and the standard deviation (SD) of each analyte from samples of the same donor across all timepoints, calculates $z = \frac{value - mean}{SD}$ for the analyte at individual timepoints, and then counts at individual timepoints how many analytes are outliers with $|z| > z_0$, where $z_0$ is a user selected cutoff value. Assuming $z$ follows a normal distribution, it is straightforward to calculate the expected rate $r$ of analytes having $|z| > z_0$ (two-sided) or having $z > z_0$ or $z < -z_0$ (one-sided). Afterwards function *outlierDetectP()* uses binomial tests to evaluate the p values for the counts of outliers at individual timepoints and applies Benjamini and Hochberg procedure to adjust the $p$ values since multiple timepoints are tested. A donor-specific abnormal timepoint is identified if the corresponding adjusted $p$ value is less than 0.05. In this study we chose $z_0 = 2.5$ and thus $r = 1.24\%$ for $|z| > 2.5$ or $r = 0.62\%$ for $z > 2.5$ or $z < -2.5$. While the $z$-score method described here can handle data with only three timepoints, Dixon's test may be a better alternative for such a small dataset.

**Time course analysis (TCA).** Function *sclongitudinalDEG()* uses the hurdle model implemented in the MAST package (https://github.com/RGLab/MAST/) to study temporal changes in longitudinal scRNA-seq data. The data is first split into subsets of individual cell types and individual participants and then analyzed independently. If the data has at least three timepoints, the function models normalized expression of each gene as a linear function of time and evaluates the slope of time and the corresponding $p$ value (likelihood ratio test). If the data has only two timepoints, the function performs DEG analysis between the two timepoints as implemented in MAST and obtains fold change and the corresponding $p$ value. Potential confounding factors (such as experimental batch, sex, age, etc.) can be specified by parameter *adjfac* which are adjusted in the analysis. Genes that are expressed in less than a certain fraction of cells (specified by parameter *mincellsexpressed*, default 0.1) are filtered out from the analysis. Obtained $p$-values are adjusted for multiple comparisons using the Benjamini and Hochberg procedure. Adjusted $p$-value <0.05 were considered significant in this study.

**Circos plots for displaying stability patterns.** PALMO has two functions to show the stability patterns of single-cell omics data. Function *genecircosPlot()* displays the CV values of features of interest in individual cell types and across individual donors based on a single data modality. Function *multimodalView()* displays the CV values of features of interest in individual cell types and across individual donors based on two independent data modalities.

**Random correlation between gene expression and gene score**
To generate the distribution of random correlation between gene expression in scRNA-seq data and gene score in scATAC-seq data, we randomly shuffled the order of reliable genes, calculated the correlations between expression of pre-shuffle genes and gene score of post-shuffle genes at the same positions, and repeated the process 1000 times. The obtained distribution of correlations provided a good estimate on the correlation between random, unrelated gene pairs, which had a 95% upper confidence bound at $R_0 = 0.399$. Any correlations below $R_0$ were no better than that between random, unrelated gene pairs and thus not statistically meaningful.

**Published single cell datasets**
We retrieved scRNA-seq data from published PBMC datasets CNP0001102[3], GSE149689[2], and GSE164378[16]. Datasets CNP0001102 and GSE164378 were from longitudinal studies. Single-cell data objects were created in Seurat v4.0.0 and cells were labeled as in the original studies. Dataset CNP0001102 consists of three healthy controls (normal), two participants infected with influenza (Flu) and five participants infected with SARS-CoV-2 (COVID-19). Dataset GSE149689 consists of four normal, five Flu, and eleven COVID-19 participants. Dataset GSE164378 dataset consists of eight participants with PBMC samples collected at three timepoints.

Mouse brain scRNA-seq data was obtained from published dataset GSE129788[36]. The dataset contains single cell RNA data from brain tissues of eight young (2–3 months) and eight old (21–23 months) mice. The dataset consists of a total 37,069 cells labeled to 25 cell types.

**TCRß repertoire dataset**
We downloaded the TCRβ sequencing data of 4 systemic sclerosis patients from GSE156980[26]. First, we merged the TCR repertoire data from the 4 patients with 3 timepoints into a single file. Second, we calculated the frequency of each unique CDR3 peptide in each patient sample as the ratio between the observed reads of the peptide to the total peptide reads in the sample. Third, we termed unique CDR3 peptides as clonotypes and labeled them from 1 to the total number of clonotypes. In total, we collected 288,597 (out of 355,024) unique clonotypes from CD4+ T cells and 11,739 (out of 14,883) from CD8+ T cells, respectively. The frequency data matrix from CD4+ or CD8+ T cells was then submitted to PALMO as input data frame.

**Differential expression gene (DEG) analysis on scRNA-seq data**
DEG analysis on datasets (CNP0001102 and GSE149689) was performed using the FindMarkers function from the Seurat package (version 4.0.0). The groups were specified using "ident.1" and "ident.2" in the function. The Benjamini and Hochberg (BH) procedure as implemented in the Seurat package was applied to adjust p-values, controlling the false discovery rate (FDR) in multiple testing. DEGs were identified if the corresponding average log2-Fold change was greater than 0.1 and the corresponding adjusted $p$ value was less than 0.05.

**Seurat differential analysis on longitudinal scRNA-seq data of a COVID19 patient**
Seurat based differential analysis was performed on the longitudinal scRNA-seq data of activated CD4+ T cells of patient COV-5 in dataset CNP0001102[3], using the function *FindMarkers()* with parameters test.use = "MAST" and logfc.threshold = 0. The groups were defined by parameters ident.1 and ident.2. For example, to capture differential genes between day 1 (D1) versus day 7 (D7) and day 13 (D13), we selected ident.1 = D1 and ident.2 = (D7 and D13). Similar approach was carried out for comparing D13 versus D1 and D7 (ident.1 = (D1 and D7) and ident.2 = D13). The significant genes were identified by adjusted $p$ value <0.05.

**Pathway enrichment analysis**
Fast Gene Set Enrichment Analysis (fgsea) was performed to identify enriched pathways among targeted genes[41]. A custom collection of gene sets that included the GO v7.2, KEGG v7.2 and Hallmark v7.2 from the Molecular Signatures Database (MSigDB, v7.2) were used as the pathway database. Genes were pre-ranked by the decreasing order of their correlation coefficients. The running sum statistics and Normalized Enrichment Scores (NES) were calculated for each comparison. The pathway enrichment p-values were adjusted using the Benjamini and Hochberg procedure and pathways with adjusted $p$-values <0.05 were considered significantly enriched. Over

representation analysis was performed using the Fisher test. For a single sample GSEA (ssGSEA), we used the GSVA v1.40 R package[27].

## Data analysis and visualization

Data analysis was performed in R, a statistical computing language (https://www.R-project.org/). Basic data visualization was performed using ggplot2 v3.3, ggpubr 0.4, and circular plots by circlize v0.4. The UMAP visualization was performed using Seurat v4.0.0. Statistical tests were performed as mentioned in each section. Multi-test correction was applied to the p-values to control the FDR using the Benjamini and Hochberg procedure and adjusted $p < 0.05$ were considered significant.

## Data availability

The processed scRNA-seq and scATAC-seq data of human PBMC samples generated in this study can be downloaded from the GEO database under accession number GSE190992. The corresponding raw sequencing data can be downloaded via authorized access from the dbGaP database under accession number phs003203.v1.p1. Complete blood count (CBC) data is provided in Supplementary Data 1a. The matching flow cytometry data of human PBMC samples is provided in Supplementary Data 1b. The matching Olink data of human plasma samples is provided in Supplementary Data 1c. Source data are provided with this paper. Independent datasets used for evaluation are publicly available and summarized in Supplementary Fig. 1b. The corresponding accession numbers are CNP0001102 (scRNA-seq data of human PBMC samples), GSE149689 (scRNA-seq data of human PBMC samples), GSE164378 (CITE-seq data of human PBMC samples), GSE129788 (scRNA-seq data of mouse brain tissues), and GSE156980 (TCRβ sequencing data of CD4+ and CD8+ non-naïve T cells of systemic sclerosis patients). Source data are provided with this paper.

## Code availability

An open-source R implementation of PALMO and R codes used in this study are available at GitHub (https://github.com/aiimmunology/PALMO). The release includes tutorials and example vignettes. PALMO can also be installed in R or RStudio as an R package in CRAN. Source code can also be found at Zenodo[42] [https://doi.org/10.5281/zenodo.7549226].

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

## Acknowledgements

We are grateful to the individuals who provided biological material for this study. We are especially thankful to all the members of the Allen Institute for Immunology and the facilities and operations teams at the Allen Institute who helped establish the productive environment in which this work was performed. We thank the study participants for their dedication to this project. We thank Drs. Gregory Lee Szeto, Emma Kuan, Claire Gustafson, Samir Rachid Zaim, and Ziyuan He for many fruitful discussions and Ms. Nina Kondza, Kathy Henderson, Muriel Ross, Kelli Burley, and Tanja Smith for blood processing. We are grateful for the leadership and support of Allan Jones, President and CEO of the Allen Institute, Allen Institute founder, Paul G. Allen, for his vision, encouragement, and support; the Human Immune System Explorer (HISE) software development team at the Allen Institute for Immunology for their support and dedication. This paper and the research behind it would not have been possible without the collaborative computational data analysis environment provided by HISE.

## Author contributions

S.V.V., A.K.S., P.J.S., T.F.B., and X.L. conceived the study. A.K.S., P.J.S. and T.F.B. designed the experiments. S.V.V., A.T and X.L. designed and developed the software package. A.K.S., A.T.H. and J.R. prepared the P.B.M.C. and plasma from whole blood. A.K.S., A.T.H. and J.R. performed the flow cytometry. A.K.S. and S.V.V. analyzed the flow cytometry data. E.S., C.L. and L.T.G. performed the scRNA-seq and the scATAC-seq experiments and pre-processing. S.V.V., A.T., Q.G. and X.L. analyzed the scRNA-seq and the scATAC-seq data. S.V.V. and X.L. analyzed the proteomics data. P.M., T.R.T., P.J.S., T.F.B., and X.L. provided direction and oversight. S.V.V. and X.L. wrote the manuscript, and all authors provided edits and comments to the manuscript.

## Competing interests

S.V.V., A.S., T.T., P.S., T.F.B., and X.L. are listed as inventors in a US patent application "Molecular Signatures For Cell Typing And Monitoring Immune Health" (application No. 63/291,234) based on this work. C.L. is currently an employee of GlaxoSmithKline. The remaining authors declare no competing interests.
