## [Transparent Peer Review File · Nature Communications]

nature portfolio

Peer Review File

A comprehensive platform for analyzing longitudinal multi-omics dataREVIEWER COMMENTS

Reviewers #1-3 (Remarks to the Author: Overall significance):

In the study "PALM: a comprehensive platform for analyzing longitudinal multi-omics data", Vasaikar et al. present a platform for analysing time series data generated from bulk and single-cell omics. The authors claimed that PALM is a comprehensive and simple-to-use platform, covering different topics in longitudinal analyses such as outlier detection, variable feature identification, inter/intra-donor variation assessment, multi-omics integration, etc. However, from my perspective, this work is far from being called a platform; instead, it is more like a patchwork of scripts by assembling different published functions/utilities into one repository without deep curations. It is not comprehensive, not simple-to-use, and importantly, not well-designed as is claimed in the manuscript.

1. PALM features the analyses of both bulk and single-cell omics data. However, several analyses of single-cell data (such as detecting features contributing towards donor variation) are achieved through pseudo-bulking the cell types. There are no dedicated functions/analyses to deal with single-cell-related problems. The other example is the quantification of intra-donor variation and the detection of outliers. While a CV-based approach can probably reflect the intra-donor variations over time in bulk data, in PALM this approach is also applied to single-cell datasets (such as the use of Z scores to determine outliers), which is dubious in terms of statistical rigour. A similar issue is in determining stable versus variable features – several thresholds are simply deployed to obtain this distinction without rigorous statistical motivation. This might be useful in an individual project to explore the data of interest, but is not sufficient for a service provided by a platform.

2. PALM is not an easy-to-use platform.

2.1. Many parameters are hard coded in PALM. The users have to adjust their data greatly to accommodate the platform, like changing their metadata column names, matching the information, etc. In addition, most functions are designed in a customized manner for the authors of the manuscript, rather than suiting potential users. One of the examples can be found in the PALM repo, <https://github.com/aifimmunology/PALM/blob/main/R/avgExpCalc.R>, where the `group.by` parameter (line 19) is strictly hard coded in the code while soft coded in the interface (line 14). The authors seem not ready for pushing PALM out as a light-weight platform.

2.2. Walking through the PALM tutorial (<https://github.com/aifimmunology/PALM#introduction>), the functions in PALM are not well integrated into an entire ecosystem. For example, after finding the donor-contributing features, there is no such a function to explore these features; instead, it is a step-by-step guide in how to write analytical codes to display and visualise them. It indeed feels like walking through a series of scripts in the code repo of a project, instead of a platform.

2.3. There are several other points that make PALM hard-to-use, including: i) The package is not directly available in CRAN or Bioconductor. ii) Metadata (e.g., the names, text format, delimiter and many others) have to be arranged in a way that is exactly as the authors' data. iii) Tutorials are filled with codes with almost no explanations in each step to guide the users. iv) There are limited flexible options to perform works at the user end. v) All utilities are function-based rather than class-based (S3/4 for example), making PALM a rough collection of functions.

3. PALM is not comprehensive in terms of tackling different kinds of tasks in longitudinal data. For example, during the intra-donor analysis, each participant is associated with several time points. What if the experimental design is not balanced, say, some participants have fewer time points? Will such a CV-based analysis still be useful? What if some time points have several technical replicates? Should these replicates be collapsed before the analysis? Or the CV analysis can be applied directly? All these are directly related to Comment 2, that is, PALM cannot accommodate different datasets from the users, being only suitable to the authors' data in a relatively strict sense. The lack of comprehensive longitudinal analysis features makes it underwhelming, as the key selling point of PALM is that this is first-of-its-kind platform to analyse longitudinal multiomics data. Importantly, the presented longitudinal analysis features are again just using previously published tools.

4. Many analyses in PALM are not novel, such as the variance decomposition, DE analyses, and circos plot. Even without novel analysis provided in PALM, the authors should at least provide a quick interface to use the external tools. For example, since the authors use the linear mixed model to assess the random effects

during variance decomposition, how about adding dedicated methods to extract the deviated donors, instead of just showing a code guide to plot these genes and locate the donors (<https://github.com/aifimmunology/PALM#plot-the-top-variables>)? I also expect that more analytical workflows will be added in PALM to handle more practical problems in longitudinal analysis.

5. The authors claim that this is a first-of-its-kind platform to analyse single cell longitudinal multiomics data. There are however various tools out there that are capable of doing this. The most notable example here is Seurat which the authors actually extensively use for both data object architecture and analysis features. Seurat offers extensive integrative approaches for multiple modalities. Amongst other options, it also offers longitudinal data analysis through for example the MAST, which is again the same as what the authors use.

Reviewer #4 (Remarks to the Author: Overall significance):

Overview summary

The authors developed a platform named PALM to conduct an integrative and comprehensive analysis using longitudinal bulk and single-cell multi-omics datasets. Applying PALM, they can perform variance decomposition, identify STATIC features, determine the abnormal dataset outliers. Overall, PALM provides a unique and comprehensive insight to understand the biological questions.

Major concerns

1. The authors demonstrate various functions within the PALM platform. However, I got quite confused what is the input for PALM. Does PALM take Seurat object as an input or does it take the raw matrices as inputs? Also, the analytical pipeline of PALM is a little bit ambiguous. Can you explain the pipeline of PALM?
2. In this manuscript, the authors provide a lot of data including 4 scRNA-seq samples with 10 different time points each. I am wondering whether we can apply the PALM platform to small datasets, like only 1 scRNA-seq sample with 2 or 3 different time points? Specifically, can the detection of abnormal timepoints function work successfully?
3. An important finding in this manuscript is STATIC identification. I wonder what is the major difference between STATIC and conserved differentially expressed genes/features across time points? Specifically, if you run an alignment algorithm like Seurat or LIGER using all different time points and different samples, and you identified the conserved differentially expressed genes for each cluster, I am wondering how many overlaps between the STATIC genes and the conserved differentially expressed genes?

Minor concerns:

1. On page 3 lines 101-105, there are lots of longitudinally variable and stable genes. I wonder what are the biological insights to determine the longitudinally variable and stable genes? Are you considering this in the developmental processes or drug treatment/perturbation processes?
2. For the scRNA-seq method part, it is a little bit unclear how many samples and how many time points were used for sequencing. Also, I am confused about why you run your scRNA-seq QC (nFeature, percent.mt, etc.) after you conduct the label transferring?
3. The description of your linear mixed model is not clear. You have considered donor, time, and cell type as random effects but not mentioned what variables you use for fixed effects. Could you provide more details?

Reviewer #5 (Remarks to the Author: Overall significance):

In their submitted work entitled "PALM: a comprehensive platform for analyzing longitudinal multi-omics data", the authors developed PALM, a platform to analyze longitudinal bulk and single-cell multi-omics data. They applied PALM to their own and public datasets to show the utility of this approach. PALM is a simple and useful tool. However, the paper suffers from several major limitations.

1. The authors applied PALM to analyze scRNA-seq and scATAC-seq datasets and their analyses were mainly focused on PBMCs. The tissue samples may have different cell compositions from blood samples, but they

were not included. Currently, huge amount of scRNA-seq and scATAC-seq data have been generated on tissue samples from many published studies, the authors are suggested showing the performance of PALM on tissue samples.

2. Likewise, the authors utilized PALM to analyze data generated on samples from healthy donors but did not show its performance on data generated from subjects with other conditions such as cancer, developmental diseases, or other genetic diseases, and adding such examples are encouraged in order to show the potentially broad application of PALM.
3. There are other existing tools that have been already developed to integrate single-cell multi-omics data (e.g., scRNA-seq and scATAC-seq data integration) and the authors are suggested comparing the performance of PALM with other tools.
4. Can PALM be used to analyze scTCR/BCR-seq data? scTCR/BCR-seq approach is commonly used in biomedical research. The same for spatial transcriptomics /proteomics data.
5. To better understand the SUV, SUS, STATIC gene lists, analyses across different tissue types, cell types (e.g., normal, inflammatory, or malignant cell types and states), and across different disease conditions are necessary.
6. The key data quality control steps warrant further check as some of the key steps of data processing (such as batch effects evaluation, correction, doublet removal, etc.) are not provided in the Methods.
7. It is unclear how PALM defines cell types/states, and cell types/states identified in PBMCs are fewer (not comprehensive) than previously reported.
8. Importantly, data analysis modules included in PALM are kind of basic. To comprehensively analyze the multi-omics data, deep profiling approach would be needed.

Response to Reviewers

In this response, original comments by editors and reviewers are presented in blue. Our detailed response is provided under the comments in black. We changed the package name from PALM to PALMO since CRAN already has a package named PALM.

Editorial assessment and review synthesis

Editor's summary and assessment

Here, the authors present PALM, a bioinformatic tool that can integrate bulk and single-cell data (including RNA-seq, proteomics, or ATAC-seq), along with clinical metrics like blood count, essentially streamlining the analysis of these varied data types. PALM is designed to integrate longitudinal data sets, to help users evaluate the stability of certain features in a dataset. As a demonstration of this workflow, the authors evaluate PBMC datasets from healthy participants or COVID-19 patients. They identify "STATIC" genes that are stable in terms of expression/chromatin profiles for each cell type, and resolve patient- and cell type-specific changes that might be linked to disease severity. Altogether, they conclude that PALM is a comprehensive platform to analyze longitudinal multi-omics data.

While the editors jointly decided to send this manuscript out to review based on the potential of the longitudinal analyses central to PALM, there were some concerns about the lack of benchmarking or limited complexity of the underlying samples, which prohibited further consideration by *Nature Methods*.

We thank the editors to get our manuscript reviewed. To demonstrate PALMO can handle simple and complex longitudinal omics datasets, we applied it to six external omics datasets of diverse complexities (**Supplementary Fig. 1c**), including one synthetic gene expression dataset, one T-cell receptor (TCR) sequencing dataset, three scRNA-seq datasets of peripheral blood mononuclear cells (PBMCs), and one scRNA-seq dataset of mouse brain tissues. These examples, in addition to those on our own complex longitudinal multi-omics data of five data modalities, provide solid evidence on PALMO usefulness to a broad audience.

The lack of well-accepted software package for longitudinal omics data makes it difficult to benchmark PALMO performance. We compared PALMO with variancePartition, tcr, and Seurat and showed PALMO either complements or surpasses these packages on longitudinal omics data.

Editorial synthesis of reviewer reports

While Reviewers #4-5 find PALM to be of potential interest to the field, they raise several concerns regarding the degree of biological insight, underpinnings of the method, and its applicability to incomplete datasets or more complex samples (e.g. from another disease context, or tissue vs. PBMC samples). Reviewers #1-3 (co-reviewers) also comment on the limited technical advance of the method over tools like Seurat, potential issues in the central longitudinal analyses, and barriers for potential users of PALM. Taken together, these points supported the initial concerns from *Nature Methods*.

We thank all reviewers for their thorough review of our manuscript and their constructive comments. We appreciate the comments by Reviewers #4-5 that our manuscript is “of potential interest to the field”. We have addressed concerns by all reviewers in the revised manuscript as detailed below. While we appreciate many constructive comments by Reviewers #1-3, we would like to point out that Seurat requires users to select two contrast groups for differential expression gene (DEG) analysis and thus is not appropriate for analyzing longitudinal data of more than two timepoints. Nevertheless, we compared PALMO and Seurat on a longitudinal scRNA-seq dataset of a COVID patient and found PALMO results showing better dynamic changes over time than Seurat results (**Supplementary Fig. 12**). We respect the decision by the editor of *Nature Methods*.

While *Nature Methods* is unable to offer a revision, *Nature Communications* would be interested in considering a revised manuscript that thoroughly demonstrates that PALM is capable of analysing single-cell datasets rigorously and specifically (Reviewers #1-3), allows users to define parameters (Reviewers #1-3), includes benchmarking to at least one alternative method (all reviewers), provides a benchmark with incomplete or smaller datasets (Reviewers #1-4), evaluates PALM on other complex datasets (Reviewer #5), and addresses technical issues with the analytical pipeline (all reviewers). Please do take into consideration that *Nature Communications* would not consider your manuscript further if your conclusions are weakened after addressing these concerns.

We thank the editor of *Nature Communications* for inviting us to submit a revised manuscript. We have addressed all concerns by the editor and the reviewers, which has strengthened not weakened our conclusions. We hope the editor and the reviewers agree with us that our manuscript is now acceptable for publication in *Nature Communications*.

- As detailed below, we have provided additional examples (**Supplementary Fig. 1c**) to demonstrate that PALMO “is capable of analyzing single-cell datasets rigorously and specifically”.
- We have made PALMO more user friendly and provided detailed instructions on how to “define parameters” in its reference manual and several examples on its usage in vignettes (<https://github.com/aifimmunology/PALMO>). PALMO has been published as an R package in CRAN and can be easily installed and executed in R or RStudio (<https://cran.r-project.org/web/packages/PALMO/index.html>).
- We have benchmarked STATIC genes against biomarkers for cell types, biomarkers for biological conditions, and highly variable genes (HVGs).
- We have benchmarked PALMO against variancePartition, tcr, and Seurat despite a lack of well-accepted software package for longitudinal omics data. PALMO either complements or surpasses these packages on longitudinal omics data.
- We have benchmarked PALMO on omics datasets of diverse complexities (**Supplementary Fig. 1c**), including “smaller” and complex datasets. More examples of PALMO usage can be found in PALMO vignettes (<https://github.com/aifimmunology/PALMO/blob/main/Vignette-PALMO.pdf>), including performance on unbalanced data, data with replicates, and data of a single donor with multiple timepoints (**Section 3.10**). We do not include these examples in the manuscript due to space limitation.
- As detailed below, we have addressed all technical issues raised by the reviewers.

Alternatively, *Communications Biology* would be interested in considering a revised manuscript that at least qualifies any concerns about user-friendliness (even if the input format is not changed), includes benchmarking to at least one alternative method, discusses the potential of PALM to analyze incomplete or smaller datasets, and clarifies the overarching analytical pipeline.

We thank the editor of *Communications Biology* for his interest in a revised manuscript. As mentioned above, we have addressed all concerns by the editor and the reviewers. We are submitting our revised manuscript for consideration to publish in *Nature Communications*.

Editorial recommendation

***Nature Methods*: Revision not invited**

Neither the conceptual advance nor advance in performance demonstrated is sufficient for publication in *Nature Methods*.

We respect the decision by the editor of *Nature Methods*.

***Nature Communications*: Major revisions with extension of the work**

Nature Communications would be interested in considering a revised manuscript that thoroughly demonstrates that PALM is capable of analyzing single-cell datasets rigorously and specifically (Reviewers #1-3), allows users to define parameters (Reviewers #1-3), includes benchmarking to at least one alternative method (all reviewers), provides a benchmark with incomplete or smaller datasets (Reviewers #1-4), evaluates PALM on other complex datasets (Reviewer #5), and addresses technical issues with the analytical pipeline (all reviewers). Please do take into consideration that *Nature Communications* would not consider your manuscript further if your conclusions are weakened after addressing these concerns.

We thank the editor of *Nature Communications* for inviting us to submit a revised manuscript. As mentioned above, we have addressed all concerns by the editor and the reviewers, which has strengthened not weakened our conclusions. We hope the editor and the reviewers agree with us that our manuscript is now acceptable for publication in *Nature Communications*.

***Communications Biology*: Major revisions**

Communications Biology would be interested in considering a revised manuscript that qualifies concerns about user-friendliness (per Reviewers #1-3), includes benchmarking to at least one alternative method (all reviewers), discusses the potential of PALM to analyze incomplete or smaller datasets (Reviewers #1-4), and clarifies the analytical pipeline (all reviewers).

We thank the editor of *Communications Biology* for his interest in a revised manuscript. As mentioned above, we have addressed all concerns by the editor and the reviewers. We are submitting our revised manuscript for consideration to publish in *Nature Communications*.

Next steps

Editorial recommendation 1:

Our top recommendation is to revise and resubmit your manuscript to *Communications Biology*. This option might be best if not all of the requested experimental revisions are possible/feasible at this time.

Editorial recommendation 2:

You may also choose to revise and resubmit your manuscript to *Nature Communications*. While we feel the additional required experiments are reasonable to achieve within a timeframe of 6 months, please keep in mind

that *Nature Communications* would not consider your manuscript further if the conclusions are weakened after addressing these concerns.

Note:

As stated on the previous page *Nature Methods* is not inviting a revision at this time. Please keep in mind that the journal will not be able to consider any appeals of their decision through Guided Open Access.

We thank all editors and reviewers for providing a thorough review on our manuscript and giving us constructive comments. We have made substantial changes to address all comments and thus significantly improved our manuscript. We are submitting our revised manuscript for consideration to publish in *Nature Communications*.

Reviewer #1-3 Comments

Editor's comments

Please note that Reviewers #1-3 are co-reviewers, so their comments are identical. These reviewers highlighted barriers to user uptake of PALM, potential issues with the quality of the analysis, and limited technical advance over methods like Seurat, prohibiting further consideration by *Nature Methods*.

We thank the reviewers for their thorough review of our manuscript and their constructive comments. Below are a few highlights of our response to their comments:

- As mentioned above, we have made PALMO more user friendly and provided detailed instructions in its reference manual and several examples on its usage in vignettes (<https://github.com/aifimmunology/PALMO>). PALMO has been published as an R package in CRAN and can be easily installed and executed in R or RStudio (<https://cran.r-project.org/web/packages/PALMO/index.html>).
- We have provided additional examples to demonstrate the usefulness of PALMO on omics datasets of diverse complexities (**Supplementary Fig. 1c**). For example, we applied PALMO to scRNA-seq data of mouse brain tissues and identified 304 STATIC genes that 1) clearly separate cell types on UMAP, 2) are mostly marker genes for cell types, and 3) are enriched for young versus old aging DEGs (**Fig. 5**), using cell types, gene markers and DEGs identified in the original study. These results reproduced our original findings from PBMC scRNA-seq data that STATIC genes are enriched for biomarkers for cell types and/or biological conditions. We anticipate many researchers may be interested in identifying STATIC genes in their own scRNA-seq datasets.
- We would like to point out that Seurat requires users to select two contrast groups for DEG analysis and thus is not appropriate for analyzing longitudinal data of more than two timepoints. Nevertheless, we compared PALMO and Seurat on a longitudinal scRNA-seq dataset of a COVID patient and found PALMO results showing better dynamic changes over time than Seurat results (**Supplementary Fig. 12**).

Remarks to the Author: Overall significance

In the study "PALM: a comprehensive platform for analyzing longitudinal multi-omics data", Vasaikar et al. present a platform for analysing time series data generated from bulk and single-cell omics. The authors claimed that PALM is a comprehensive and simple-to-use platform, covering different topics in longitudinal analyses such as outlier detection, variable feature identification, inter/intra-donor variation assessment, multi-omics integration, etc. However, from my perspective, this work is far from being called a platform; instead, it is more like a patchwork of scripts by assembling different published functions/utilities into one repository without deep curations. It is not comprehensive, not simple-to-use, and importantly, not well-designed as is claimed in the manuscript.

We thank the reviewers for their constructive comments. To address their platform concern, the updated package creates a PALMO S4 object from each longitudinal omics dataset and the corresponding metadata and uses it as the starting point for downstream analyses (lines 715-737, pages 16-17), including those to be developed in the future.

PALMO analyzes longitudinal omics data from multiple perspectives, which is necessary to address diverse research interest and/or study design in longitudinal studies but may appear to be “patchwork” to the reviewers. More detailed response to their “patchwork” comment is provided under their Comment 3. We modified our summary of PALMO and removed the phrases that the reviewers expressed objection.

1. PALM features the analyses of both bulk and single-cell omics data. However, several analyses of single-cell data (such as detecting features contributing towards donor variation) are achieved through pseudo-bulking the cell types. There are no dedicated functions/analyses to deal with single-cell-related problems. The other example is the quantification of intra-donor variation and the detection of outliers. While a CV-based approach can probably reflect the intra-donor variations over time in bulk data, in PALM this approach is also applied to single-cell datasets (such as the use of Z scores to determine outliers), which is dubious in terms of statistical rigour. A similar issue is in determining stable versus variable features – several thresholds are simply deployed to obtain this distinction without rigorous statistical motivation. This might be useful in an individual project to explore the data of interest, but is not sufficient for a service provided by a platform.

Justification of PALM’s analytical pipeline would be necessary for further consideration at *Communications Biology*. However, for further consideration at *Nature Communications*, it would also be essential to thoroughly demonstrate that PALM’s model is capable of analysing single-cell datasets rigorously and specifically; for instance, with ground truth-based benchmarking.

The reviewers are correct that we apply pseudo-bulking to single-cell data in three modules (**Fig. 1b**). We have devoted one paragraph in Discussion (lines 437-448, page 10) to discuss our rational behind our approach. The main points include:

- Recent literature revealed that single-cell methods are NOT better than pseudo-bulk methods if generalized linear mixed model (GLMM) is not applied to account for interdependency of cells of same sample in cross-sectional studies. The interdependency issue is even more complicated in longitudinal studies than cross-sectional studies.
 - Dal Molin et al., *Front. Genet.* 8, 62 (2017).
 - Squir et al., *Nat. Commun.* 12, 5692 (2021).
 - Zimmerman et al., *Nat. Commun.* 12, 738 (2021).
- Longitudinal single-cell omics data is typically large. For example, we collected high quality scRNA-seq data of 472,464 cells in our dataset. It is computationally challenging to apply GLMM to data of such a volume even with cloud-based computing.
- We have adopted the pseudo-bulk approach in three analytical modules as a practical compromise.
- In the module of time course analysis (TCA), we subset scRNA-seq data by individual cell types and individual participants, which allows us to apply the single-cell hurdle model to evaluate transcriptomic changes over time (lines 801-814, page 18).

We have provided multiple examples to demonstrate that PALMO can analyze single-cell datasets “rigorously and specifically” (**Figs. 2,4,5,7**) and “with ground truth-based benchmarking” (**Figs. 4,5**).

- In **Fig. 4c-f** and **Fig. 5b**, we generated UMAP using STATIC genes (sUMAP) on five scRNA-seq datasets of two different sample types. The cells are labeled as in the originally studies. In all cases, we showed that STATIC genes (220 for human PBMCs and 304 for mouse brain tissues) were able to separate cell types well on sUMAP.
- In **Fig. 5c**, we showed 299 out of the 304 STATIC genes were identified as gene markers for the corresponding cell types in the original study.
- In **Fig. 4h,i** and **Fig. 5d**, we showed STATIC genes were enriched for biological biomarkers either using the same methods described in the original studies (**Fig. 4h,i**) or as identified in the original study (**Fig. 5d**).
- In **Fig. 4g**, we showed that STATIC genes were significantly more correlated between expression in scRNA-seq data and gene score in scATAC-seq data than highly variable genes (HVGs), which are widely used in dimension reduction on scRNA-seq data.

We have provided more technical details on PALMO and discussed the selection of various thresholds (lines 708-820, pages 16-18). We have also demonstrated how to select thresholds based on individual datasets and pass them as parameters to various PALMO functions in vignettes (<https://github.com/aifimmunology/PALMO>).

2. PALM is not an easy-to-use platform. 2.1. Many parameters are hard coded in PALM. The users have to adjust their data greatly to accommodate the platform, like changing their metadata column names, matching the information, etc. In addition, most functions are designed in a customized manner for the authors of the manuscript, rather than suiting potential users. One of the examples can be found in the PALM repo, <https://github.com/aifimmunology/PALM/blob/main/R/avgExpCalc.R>, where the group.by parameter (line 19) is strictly hard coded in the code while soft coded in the interface (line 14). The authors seem not ready for pushing PALM out as a light-weight platform.

For further consideration at *Nature Communications*, we would expect you to address this concern by allowing the potential users to define relevant parameters. *Communications Biology* would also encourage this change, though this point could also be addressed by qualifying any claims of user-friendliness throughout the text.

We took this critique very seriously and have substantially improved PALMO. In the updated version, users can pass column names in their metadata as parameters to various PALMO functions and get their data analyzed without much manipulation. As mentioned above, we have made PALMO more user friendly and provided detailed instructions in its reference manual and several examples on its usage in vignettes (<https://github.com/aifimmunology/PALMO>). PALMO has been published as an R package in CRAN and can be easily installed and executed in R or RStudio. As an example, below is a screenshot on the updated *avgExpCalc* function in PALMO reference manual.

avgExpCalc	avgExpCalc Function
------------	----------------------------

Description

This function allows you to calculate average gene expression on log-normalized data by group defined by user. This function uses Seurat function AverageExpression (<https://satijalab.org/seurat/reference/averageexpression>).

Usage

```
avgExpCalc(data_object, assay = "RNA", group_column)
```

4 *checkReplicates*

Arguments

data_object	Input PALMO S4 object. Contains annotation table and expression matrix or data frame. Rows represent gene/proteins column represents participant samples (same as annotation table Sample column)
assay	Single cell data Assay type ('RNA', 'SCT'). Default 'RNA'
group_column	Calculate average expression by given group like 'celltype' or 'cluster'

Value

PALMO object with avg expression

Examples

```
## Not run:
palmo_obj=avgExpCalc(data_object=palmo_obj, assay='RNA',
group_column='celltype')

## End(Not run)
```

2.2. Walking through the PALM tutorial (<https://github.com/aifimmunology/PALM#introduction>), the functions in PALM are not well integrated into an entire ecosystem. For example, after finding the donor-contributing features, there is no such a function to explore these features; instead, it is a step-by-step guide in how to write

analytical codes to display and visualise them. It indeed feels like walking through a series of scripts in the code repo of a project, instead of a platform.

We have made the following changes to address these comments: First, we have created a PALMO S4 object system to store input data, input parameters and output results in a single ecosystem that allows easy retrieval and interpretation (lines 715-737, pages 16-17; **Section 3.9** of PALMO vignettes). Second, we have added multiple functions to facilitate the display of PALMO output results, including *dimUMAPPlot()*, *genecircosPlot()*, *gene_featureplot()*, *multimodalView()*, and *variancefeaturePlot()*. On the specific example given by the reviewers, users can use the function *variancefeaturePlot()* to visualize the donor- and time-contributing features (**Section 3.1.6** of PALMO vignettes).

PALMO vignettes: <https://raw.githubusercontent.com/aifimmunology/PALMO/main/Vignette-PALMO.pdf>

2.3. There are several other points that make PALM hard-to-use, including: i) The package is not directly available in CRAN or Bioconductor. ii) Metadata (e.g., the names, text format, delimiter and many others) have to be arranged in a way that is exactly as the authors' data. iii) Tutorials are filled with codes with almost no explanations in each step to guide the users. iv) There are limited flexible options to perform works at the user end. v) All utilities are function-based rather than class-based (S3/4 for example), making PALM a rough collection of functions.

Both *Nature Communications* and *Communications Biology* would strongly recommend that the package be made available in CRAN or Bioconductor.

We have made the following changes to address these comments.

- i) As mentioned above, PALMO has been published as an R package in CRAN and can be easily installed and executed in R or RStudio (<https://cran.r-project.org/web/packages/PALMO/index.html>).
- ii) In the updated PALMO, users can use function *annotateMetadata()* to pass column names in their metadata to designated variables (*sample_column*, *donor_column*, and *time_column*) of the PALMO object for longitudinal analysis. See below for an example.

```
#Assign Sample, PTID and Time parameters
palmo_obj<- annotateMetadata(data_object=palmo_obj,
                             sample_column= "Sample", donor_column= "PTID",
                             time_column= "Time")
```

- iii) As mentioned above, we have made PALMO more user friendly and provided detailed instructions in its reference manual and several examples on its usage in vignettes (<https://github.com/aifimmunology/PALMO>), including step-by-step procedures for users to understand each step, QC or tweak the parameters as necessary.
- iv) Following the reviewers' suggestion, we have parameterized variables in PALMO functions which makes them more flexible for analyzing diverse datasets. For example, we used the module SPECT to analyze a scRNA-seq dataset from a cross-sectional mouse brain study (**Fig. 5, Section 3.7** of PALMO vignettes). We treated data from the eight samples of each age group as repeated measurements for the group, just like repeated measurements at different timepoints in a longitudinal study. Since SPECT does not utilize the ordering of timepoints, its usage to the data is justified. More specifically, we used "donor_column="Age_group"" to mimic samples of same age group as samples of same donor in a longitudinal study and "time_column="Subject_id"" to mimic samples of individual mice as samples of individual timepoints. As shown in **Fig. 5**, SPECT identified 304 STATIC genes that significantly overlap with gene markers for cell types and young versus old aging biomarkers. Below is the command that enables the analysis.

```
#Assign Sample, PTID and Time parameters
palmo_obj <- annotateMetadata(data_object=palmo_obj,
                             sample_column="Sample",
                             donor_column="Age_group",
                             time_column="Subject_id")
```

- v) We have made the following changes to address this comment: First, as mentioned above, we have created a PALMO S4 object system to store input data, input parameters and output results in a single

ecosystem and to serve as the starting point for downstream analyses (lines 715-737, pages 16-17; **Section 3.9** of PALMO vignettes). Second, we have made the function-based modules compatible with S4 class. Overall, the S4 object system has made it easy to access data and to perform longitudinal analysis and has enhanced PALMO capability.

3. PALM is not comprehensive in terms of tackling different kinds of tasks in longitudinal data. For example, during the intra-donor analysis, each participant is associated with several time points. What if the experimental design is not balanced, say, some participants have fewer time points? Will such a CV-based analysis still be useful? What if some time points have several technical replicates? Should these replicates be collapsed before the analysis? Or the CV analysis can be applied directly? All these are directly related to Comment 2, that is, PALM cannot accommodate different datasets from the users, being only suitable to the authors' data in a relatively strict sense. The lack of comprehensive longitudinal analysis features makes it underwhelming, as the key selling point of PALM is that this is first-of-its-kind platform to analyse longitudinal multiomics data. Importantly, the presented longitudinal analysis features are again just using previously published tools. **This point was also raised by Reviewer #4. Performing an actual analysis to demonstrate this point using smaller datasets (e.g. with some samples missing time points) would be essential for further consideration at Nature Communications and Communications Biology.**

We have provided more details in **Methods** to address the questions raised by the reviewers on CV-based modules (CVP and SPECT; lines 754-782, pages 17-18).

- Both modules first calculate CV of individual features on samples of same participant and then compare CV values of same features across participants. Thus, these modules are insensitive to unbalanced data as long as each participant has at least two time points. We believe these CV-based modules are still useful for longitudinal omics data of a few timepoints.
- PALMO takes the median values on technical replicates if they exist at some timepoints of some participants, using function *checkReplicates()*. To show PALMO can handle datasets with replicates, we applied it to a PBMC scRNA-seq dataset (GSE150861) of severe COVID-19 patients from Guo et al. (2020). The dataset contains two donors, one of whom had two replicates on both day 1 (D1) and day 5 (D5) post infection. The results are presented in **Section 3.10.3** of PALMO vignettes (<https://github.com/aifimmunology/PALMO/blob/main/Vignette-PALMO.pdf>).

We respectfully disagree with the reviewers that PALMO is “just using previously published tools”. Among the five modules in PALMO, we believe they are either novel (SPECT and ODA) or significantly improved over existing tools (TCA, CVP and VDA). Together these five modules provide unique insights on longitudinal omics data from multiple perspectives.

- We are not aware of any published tools that analyze longitudinal single-cell omics data in a similar way as SPECT (stability pattern evaluation across cell types). We demonstrated on scRNA-seq data of two sample types how SPECT identified 220 STATIC genes from human PBMCs and 304 STATIC genes from mouse brain tissues, both of which separated cell types well on UMAP and were enriched for biomarkers for cell types and/or biological conditions (**Figs. 4,5**).
- Many outlier detection methods in the literature can detect whether a specific feature is an outlier at a specific timepoint. We are not aware of any published tools that assess whether a sample in a longitudinal omics study is an outlier or not based on how many features are outliers, which is the functionality of ODA (outlier detection analysis).
- MAST is a well-known tool for DEG analysis on cross-sectional scRNA-seq data. We extended it to analyze longitudinal scRNA-seq data in TCA (time course analysis). As mentioned above, we would like to point out that Seurat requires users to select two contrast groups for DEG analysis and thus is not appropriate for analyzing longitudinal data of more than two timepoints.
- While CV calculation is widely used on omics data of multiple samples, we apply it to longitudinal omics data and take the extra step to identify consistently stable or variable features among participants in CVP (coefficient of variation profiling).
- VDA (variance decomposition analysis) and variancePartition both apply generalized linear mixed model (GLMM) to perform variance decomposition. VDA can handle missing data, which is inevitable in longitudinal omics data, but variancePartition cannot.

As mentioned above, we have benchmarked PALMO on omics datasets of diverse complexities (**Supplementary Fig. 1c**), including “smaller” and complex datasets. More examples of PALMO usage can be found in PALMO vignettes (<https://github.com/aifimmunology/PALMO/blob/main/Vignette-PALMO.pdf>), including performance on unbalanced data, data with replicates, and data of a single donor with multiple timepoints (**Section 3.10**). We do not include these examples in the manuscript due to space limitation.

4. Many analyses in PALM are not novel, such as the variance decomposition, DE analyses, and circos plot. Even without novel analysis provided in PALM, the authors should at least provide a quick interface to use the external tools. For example, since the authors use the linear mixed model to assess the random effects during variance decomposition, how about adding dedicated methods to extract the deviated donors, instead of just showing a code guide to plot these genes and locate the donors (<https://github.com/aifimmunology/PALM#plot-the-top-variables>)? I also expect that more analytical workflows will be added in PALM to handle more practical problems in longitudinal analysis.

We believe we have adequately addressed the “not novel” concern under their Comment 3. As mentioned above, we have added multiple functions to facilitate the display of PALMO output results. The idea to extract “deviated donors” is interesting but likely requires careful considerations on how to define such “deviated donors”. For example, how far and how often (across all or most timepoints) should a deviated donor separate from other donors, by what molecular features, and by what statistical criterion? We would like to leave such a functionality to future work. We admit it is not feasible for PALMO, or any single platform, to provide all necessary analytical modules for longitudinal omics data. We plan to add “more analytical workflows” to PALMO so that it can handle more and more diverse research interest in the future.

5. The authors claim that this is a first-of-its-kind platform to analyse single cell longitudinal multiomics data. There are however various tools out there that are capable of doing this. The most notable example here is Seurat which the authors actually extensively use for both data object architecture and analysis features. Seurat offers extensive integrative approaches for multiple modalities. Amongst other options, it also offers longitudinal data analysis through for example the MAST, which is again the same as what the authors use. **This point was also hinted at by Reviewers #4-5. Please benchmark PALM to at least one other existing method, for further consideration at *Nature Communications* and *Communications Biology*.**

As mentioned above, the lack of well-accepted software package for longitudinal omics data makes it difficult to benchmark PALMO performance. We compared PALMO with variancePartition, tcr, and Seurat and showed PALMO either complements or surpasses these packages on longitudinal omics data.

Seurat is a well-known platform for single-cell omics data, including modules for cell labeling, co-embedding of scRNA-seq and scATAC-seq data, and DEG analysis for cross-sectional scRNA-seq data, etc. As mentioned above, we would like to point out that Seurat requires users to select two contrast groups for DEG analysis and thus is not appropriate for analyzing longitudinal data of more than two timepoints. Nevertheless, we compared PALMO and Seurat on a longitudinal scRNA-seq dataset of a COVID patient and found PALMO results showing better dynamic changes over time than Seurat results (**Supplementary Fig. 12**).

Reviewer #4 Comments

Editor's comments

This reviewer finds PALM to be a valuable tool for evaluating longitudinal datasets, but highlights the need for clearer descriptions of the underlying method (see Major Concern #1), whether it is applicable to smaller datasets, and its performance relative to other methods.

We thank the reviewer for his/her thorough review of our manuscript and constructive comments. We are glad that the reviewer finds PALMO “to be a valuable tool”. As detailed below, we have addressed all his/her comments.

Remarks to the Author: Overall significance

Overview summary

The authors developed a platform named PALM to conduct an integrative and comprehensive analysis using longitudinal bulk and single-cell multi-omics datasets. Applying PALM, they can perform variance decomposition, identify STATIC features, determine the abnormal dataset outliers. Overall, PALM provides a unique and comprehensive insight to understand the biological questions.

We thank the reviewer for his/her very positive comment that PALMO “provides a unique and comprehensive insight to understand the biological questions”.

Major concerns

1. The authors demonstrate various functions within the PALM platform. However, I got quite confused what is the input for PALM. Does PALM take Seurat object as an input or does it take the raw matrices as inputs? Also, the analytical pipeline of PALM is a little bit ambiguous. Can you explain the pipeline of PALM?

For the sake of reproducibility, please be sure to clarify the underlying PALM pipeline. Please note that the Methods section in Nature Portfolio journals does not have a strict word limit.

To address this comment, we have added a flowchart (**Fig. 1b**) to provide an overview of the five modules in PALMO and substantially extended the description of technical details of PALMO in **Methods** (lines 708-820, page 16-18). PALMO takes as inputs a data frame of metadata describing samples and either a data frame of longitudinal data (bulk or single-cell) or a Seurat object (single-cell) to create a PALMO S4 object: see lines 722-728, page 16 for more details.

2. In this manuscript, the authors provide a lot of data including 4 scRNA-seq samples with 10 different time points each. I am wondering whether we can apply the PALM platform to small datasets, like only 1 scRNA-seq sample with 2 or 3 different time points? Specifically, can the detection of abnormal timepoints function work successfully?

This point was also raised by Reviewers #1-3. Performing an actual analysis to demonstrate this point would be essential for further consideration at Nature Communications and Communications Biology.

As mentioned above, we have benchmarked PALMO on omics datasets of diverse complexities (**Supplementary Fig. 1c**), including small and complex datasets. More examples of PALMO usage can be found in PALMO vignettes (<https://github.com/aifimmunology/PALMO/blob/main/Vignette-PALMO.pdf>), including performance on unbalanced data (**Section 3.10.2**), data with replicates (1 donor with 2 samples at each of 2 timepoints, 1 donor with 1 sample at each of 3 timepoints, **Section 3.10.4**), and data of a single donor with six timepoints (**Section 3.10.5**). We do not include these examples in the manuscript due to space limitation.

We have multiple examples to cover the small-dataset scenario described by the reviewer, including the example in **Section 3.10.4** of PALMO vignettes and the four examples in **Fig. 7a-d** and **Supplementary Fig. 12d**. PALMO worked successfully on all these examples.

Without external information to define the normal timepoints, the concept of abnormal timepoints may only apply to data of at least three timepoints. The statistical method implemented in ODA can be used to detect abnormal timepoints on data of three timepoints, but it may not be the best method for the task. It is better to use Dixon test to detect outlier features on data of three timepoints instead of the z-score method implemented in ODA. We add a sentence in **Methods** to reflect this detail: “While the z-score method described here can handle data with only three timepoints, Dixon’s test may be a better alternative for such a small dataset.” (Lines 798-800, page 18)

3. An important finding in this manuscript is STATIC identification. I wonder what is the major difference between STATIC and conserved differentially expressed genes/features across time points? Specifically, if you run an alignment algorithm like Seurat or LIGER using all different time points and different samples, and you identified the conserved differentially expressed genes for each cluster, I am wondering how many overlaps between the STATIC genes and the conserved differentially expressed genes?

The need for further benchmarking is also echoed by Reviewer #5. Please benchmark PALM to at least one other method for further consideration at *Nature Communications* and *Communications Biology*.

We thank the reviewer for recognizing the identification of STATIC genes as “important finding”. The reviewer raised a very interesting question on the overlap between STATIC genes and conserved DEGs for each cluster, which we interpret as DEGs for cell types. To answer the question, we applied SPECT to scRNA-seq data of mouse brain tissues and identified 304 STATIC genes, of which 299 were identified as gene markers for the corresponding cell types in the original study (Fig. 5c). While this overlap is rather striking, STATIC genes and DEGs are identified by completely different methods and may carry different biological insights. We explore some interesting properties of STATIC genes in Figs. 4,5. We expect to learn more on this set of genes in the future.

As mentioned above, we have benchmarked STATIC genes as input features for UMAP, against gene markers for cell types, and against biomarkers for biological conditions (Figs. 4,5). We have also compared STATIC genes with highly variable genes (HVGs) and found STATIC genes had stronger correlation between gene expression in scRNA-seq data and gene score in scATAC-seq data than HVGs (Figs. 4g). Furthermore, we have compared PALMO with variancePartition, tcr, and Seurat and showed PALMO either complements or surpasses these packages on longitudinal omics data.

Minor concerns:

1. On page 3 lines 101-105, there are lots of longitudinally variable and stable genes. I wonder what are the biological insights to determine the longitudinally variable and stable genes? Are you considering this in the developmental processes or drug treatment/perturbation processes?

Please elaborate on the underlying biological insights in the Discussion, for further consideration at *Nature Communications* and *Communications Biology*.

We thank the reviewer for answering this interesting question. We have added a paragraph in Discussion to address biological insights on variable and stable plasma proteins (lines 429-436, page 10), which is copied below. We have also added a sentence to point out this biological application in Results: “These stable proteins may be interesting biomarker candidates if they change under some disease conditions.” (Lines 206-207, page 5)

“Plasma proteins are often targeted as disease biomarkers, thus understanding their temporal stability is of particular interest. Conceptually, highly variable proteins are poor biomarker candidates since their values likely have very high sampling variations. The rather moderate CV values of the most variable proteins in our study suggest sampling variations are not a big concern on these proteins. The small CV values of the most stable proteins, on the other hand, indicate they do not change much under normal, healthy conditions. So, if they ever change under some disease conditions, they should be closely explored as potential biomarkers.”

2. For the scRNA-seq method part, it is a little bit unclear how many samples and how many time points were used for sequencing. Also, I am confused about why you run your scRNA-seq QC (nFeature, percent.mt, etc.) after you conduct the label transferring?

It would be essential to address this concern for further consideration at *Nature Communications*, preferably with additional analyses or test cases.

We realize our original presentation was not clear on these details. We have enclosed a table to describe available data for each donor and at each timepoint (Supplementary Fig. 1a). We have also cleaned up the description on technical details in **Methods** (lines 591-616, pages 13-14). Briefly, we have an automated processing pipeline for scRNA-seq data, which contains the labeling step and is executed on individual batches of scRNA-seq data separately. Before analyzing scRNA-seq data in a study, we merge all batches of its data together, calculate QC merits (nFeature, percent.mt, etc.), filter out cells of low quality, normalize the data, remove doublets, remove poorly labeled cells, and check for possible batch effects. Afterwards we carry out downstream analysis for biological insights. In short, cell labeling is performed at batch-level and QC assessment is performed later at study-level.

3. The description of your linear mixed model is not clear. You have considered donor, time, and cell type as random effects but not mentioned what variables you use for fixed effects. Could you provide more details?

We have provided more details on the linear mixed model in **Methods** (lines 738-753, page 17). The code allows for the inclusion of fixed effects for some variables if such effects are expected (see a screenshot on *lmeVariance*). We did not include any fixed effects in our examples since they are not expected on the specific datasets.

<code>lmeVariance</code>	lmeVariance Function
-----------------------------

Description

This function allows you to calculate inter-donor variation between participants over longitudinal timepoints. It uses linear mixed model to calculate variance contribution from each given feature list.

Usage

```
lmeVariance(  
  data_object,  
  featureSet,  
  fixed_effect_var = NULL,  
  meanThreshold = NULL,  
  selectedFeatures = NULL,  
  NA_to_zero = FALSE,  
  cl = 2,  
  lmer_control = FALSE,  
  fileName = NULL,  
  filePATH = NULL  
)
```

Arguments

<code>data_object</code>	Input PALMO S4 object. It contains annotation information and expression data from Bulk or single cell data.
<code>featureSet</code>	Variance analysis carried out for the feature set provided such as <code>c('PTID', 'Time', 'Sex')</code>
<code>fixed_effect_var</code>	Fixed effect variables. In linear mixed model <code>fixed_effect_var</code> included as fixed effect variables and variance contribution obtained by adding them as random variables
<code>meanThreshold</code>	Average expression threshold to filter lowly expressed genes/features Default is 0

Reviewer #5 comments

Editor's comments

This reviewer echoes several concerns from Reviewers #1-4, and also emphasizes the need for further demonstration of how PALM could be applied to complex data types (like tissue samples).

We thank the reviewer for his/her thorough review of our manuscript and constructive comments. As discussed above, we have applied PALMO to scRNA-seq data of mouse brain tissues (GSE129788, Ximerakis et al (2019)) and identified 304 STATIC genes that 1) clearly separate cell types on UMAP, 2) are mostly marker genes for cell types, and 3) are enriched for young versus old aging DEGs (**Fig. 5**), using cell types, gene markers and DEGs identified in the original study. As detailed below, we have addressed all his/her comments.

Remarks to the Author: Overall significance

In their submitted work entitled "PALM: a comprehensive platform for analyzing longitudinal multi-omics data", the authors developed PALM, a platform to analyze longitudinal bulk and single-cell multi-omics data. They applied PALM to their own and public datasets to show the utility of this approach. PALM is a simple and useful tool. However, the paper suffers from several major limitations.

We thank the reviewer for his/her comment that PALMO is "a simple and useful tool". We have addressed all his/her comments below.

1. The authors applied PALM to analyze scRNA-seq and scATAC-seq datasets and their analyses were mainly focused on PBMCs. The tissue samples may have different cell compositions from blood samples, but they were not included. Currently, huge amounts of scRNA-seq and scATAC-seq data have been generated on tissue samples from many published studies, the authors are suggested showing the performance of PALM on tissue samples.

This point would be necessary for further consideration at *Nature Communications*, but should be mentioned as a limitation for *Communications Biology*.

As mentioned above, we have applied PALMO to scRNA-seq data of mouse brain tissues (GSE129788, Ximerakis et al (2019)) and identified 304 STATIC genes that 1) clearly separate cell types on UMAP, 2) are mostly marker genes for cell types, and 3) are enriched for young versus old aging DEGs (**Fig. 5**), using cell types, gene markers and DEGs identified in the original study. These results reproduced our original findings from PBMC scRNA-seq data that STATIC genes are enriched for biomarkers for cell types and/or biological conditions.

2. Likewise, the authors utilized PALM to analyze data generated on samples from healthy donors but did not show its performance on data generated from subjects with other conditions such as cancer, developmental diseases, or other genetic diseases, and adding such examples are encouraged in order to show the potentially broad application of PALM.

Given that SARS-CoV-2 datasets were already incorporated into the manuscript, this point would not be necessary for further consideration at *Communications Biology*. However, for further consideration at *Nature Communications*, we would expect your revision to provide additional applications of PALM to datasets from other diseases.

We have added several datasets to demonstrate PALMO performance (**Supplementary Fig. 1b**), including a TCR sequencing dataset of systemic sclerosis patients and a scRNA-seq dataset of mouse brain tissues to study aging.

3. There are other existing tools that have been already developed to integrate single-cell multi-omics data (e.g., scRNA-seq and scATAC-seq data integration) and the authors are suggested comparing the performance of PALM with other tools.

The need for further benchmarking is also echoed by other reviewers. Please benchmark PALM to at least one other method for further consideration at *Nature Communications* and *Communications Biology*.

As mentioned above, the lack of well-accepted software package for longitudinal omics data makes it difficult to benchmark PALMO performance. We have compared PALMO with variancePartition, tcr, and Seurat and showed PALMO either complements or surpasses these packages on longitudinal omics data.

Seurat is a well-known platform for single-cell omics data, including modules for cell labeling, co-embedding of scRNA-seq and scATAC-seq data, and DEG analysis for cross-sectional scRNA-seq data, etc. There are major differences between Seurat and PALMO on scRNA-seq data and scATAC-data integration. The former is a sophisticated approach to co-embed cells from both data modalities in same UMAP space and thus enable cell labeling on scATAC-seq data based on scRNA-seq reference. The latter uses circos plots to display stability patterns of both data modalities across cell types and participants. We have deemphasized the importance of integrated display in the revised manuscript.

4. Can PALM be used to analyze scTCR/BCR-seq data? scTCR/BCR-seq approach is commonly used in biomedical research. The same for spatial transcriptomics /proteomics data.

We would strongly recommend you to include such benchmarking for further consideration at *Nature Communications*. In contrast, for *Communications Biology*, while direct benchmarking would not be necessary for scTCR- or scBCR-seq data, we would ask that you comment on this point in the Discussion.

We have added a TCR sequencing dataset of systemic sclerosis patients in benchmarking PALMO (**Supplementary Fig. 6**). In the analysis, PALMO treats clonotype frequency as continuous variable and examines the data at clonotype-level. On the contrary, tcr either provides sample-level views on the whole repertoires or treats clonotype data as binary (present or absent). PALMO obtained results that are potentially interesting

but not reported in the original study using tcr (lines 190-195, page 5). We believe PALMO complements TCR specific tools (such as tcr) on TCR data.

We believe PALMO can be applied to spatial transcriptomics or spatial proteomics data. But it is beyond the scope of this manuscript to demonstrate such use cases.

5. To better understand the SUV, SUS, STATIC gene lists, analyses across different tissue types, cell types (e.g., normal, inflammatory, or malignant cell types and states), and across different disease conditions are necessary. **This point would be necessary for further consideration at *Nature Communications*, but (as noted in Point #2) not *Communications Biology*.**

As mentioned above, we have applied PALMO to scRNA-seq data of mouse brain tissues (GSE129788, Ximerakis et al (2019)) and identified 304 STATIC genes that 1) clearly separate cell types on UMAP, 2) are mostly marker genes for cell types, and 3) are enriched for young versus old aging DEGs (**Fig. 5**), using cell types, gene markers and DEGs identified in the original study. These results reproduced our original findings from PBMC scRNA-seq data that STATIC genes are enriched for biomarkers for cell types and/or biological conditions.

Additional information on SUV, SUS, and VATIC genes from mouse brain tissues can be found in **Section 3.7.8** of PALMO vignettes.

6. The key data quality control steps warrant further check as some of the key steps of data processing (such as batch effects evaluation, correction, doublet removal, etc.) are not provided in the Methods.

We realize our original presentation was not clear on these details. As mentioned above, we have cleaned up the description on technical details in **Methods** (lines 591-616, pages 13-14). Briefly, we have an automated processing pipeline for scRNA-seq data, which contains the labeling step and is executed on individual batches of scRNA-seq data separately. Before analyzing scRNA-seq data in a study, we merge all batches of its data together, calculate QC merits (nFeature, percent.mt, etc.), filter out cells of low quality, normalize the data, remove doublets, remove poorly labeled cells, and check for possible batch effects. Afterwards we carry out downstream analysis for biological insights.

7. It is unclear how PALM defines cell types/states, and cell types/states identified in PBMCs are fewer (not comprehensive) than previously reported.

Cells in our scRNA-seq data were labeled using Seraut V2. We kept 19 cell types for PALMO analysis after filtering out low frequent cell types (lines 99-108, page 3; lines 591-616, pages 13-14). Cells in our scATAC-seq data were labeled using genescore matrix as implemented in ArchR (Granja et al., Nat. Genet. 53, 403–411 (2021)). We kept 14 cell types for PALMO analysis after filtering out poorly labeled and low frequent cell types (lines 109-117, page 3; lines 696-706, pages 16). Cells in public scRNA-seq datasets were labeled as in the original studies.

8. Importantly, data analysis modules included in PALM are kind of basic. To comprehensively analyze the multi-omics data, deep profiling approach would be needed.

Please see our response to Comment 3 by Reviewers #1-3. As mentioned there, among the five modules in PALMO, we believe they are either novel (SPECT and ODA) or significantly improved over existing tools (TCA, CVP and VDA). Together these five modules provide unique insights on longitudinal omics data from multiple perspectives.

We hope the reviewer finds the identification of STATIC genes novel and interesting. We admit it is not feasible for PALMO, or any single platform, to provide all necessary analytical modules for longitudinal omics data. We plan to add more modules to PALMO in the future to cover more and more diverse research interest.

REVIEWERS' COMMENTS:

Reviewer #1 (Remarks to the Author: Overall significance):

In the revised manuscript, the authors have re-structured the software ecosystem of PALMO, and enriched their tutorials with dedicated functions and explanations. Many analyses and results have been optimized, and the authors have included additional benchmarks. Some analytical pipelines are not improved, but the authors have discussed the potential pros and cons using their approaches. Overall, this work (and the tool) is significantly improved compared to the previous one, and now provides a useful toolbox for the community. Because of these improvements, together with the decision to transfer to Nature Communications, we now believe that this work is suitable for publication.

Reviewer #2 (Remarks to the Author: Overall significance): Please note that this referee co-reviewed with Reviewer #1

In the revised manuscript, the authors have re-structured the software ecosystem of PALMO, and enriched their tutorials with dedicated functions and explanations. Many analyses and results have been optimized, and the authors have included additional benchmarks. Some analytical pipelines are not improved, but the authors have discussed the potential pros and cons using their approaches. Overall, this work (and the tool) is significantly improved compared to the previous one, and now provides a useful toolbox for the community. Because of these improvements, together with the decision to transfer to Nature Communications, we now believe that this work is suitable for publication.

Reviewer #4 (Remarks to the Author: Overall significance):

The authors have fully addressed my concerns. I do not have further revisions to suggest at this moment.

Response to Reviewers

In this response, original comments by reviewers are presented in blue. Our detailed response is provided under the comments in black.

REVIEWERS' COMMENTS:

Reviewer #1 (Remarks to the Author: Overall significance):

In the revised manuscript, the authors have re-structured the software ecosystem of PALMO, and enriched their tutorials with dedicated functions and explanations. Many analyses and results have been optimized, and the authors have included additional benchmarks. Some analytical pipelines are not improved, but the authors have discussed the potential pros and cons using their approaches. Overall, this work (and the tool) is significantly improved compared to the previous one, and now provides a useful toolbox for the community. Because of these improvements, together with the decision to transfer to Nature Communications, we now believe that this work is suitable for publication.

We thank the reviewer for helping us improve our manuscript and for recommending publication in *Nature Communications*. The reviewer has no more suggestions or comments for us to address.

Reviewer #2 (Remarks to the Author: Overall significance): Please note that this referee co-reviewed with Reviewer #1

In the revised manuscript, the authors have re-structured the software ecosystem of PALMO, and enriched their tutorials with dedicated functions and explanations. Many analyses and results have been optimized, and the authors have included additional benchmarks. Some analytical pipelines are not improved, but the authors have discussed the potential pros and cons using their approaches. Overall, this work (and the tool) is significantly improved compared to the previous one, and now provides a useful toolbox for the community. Because of these improvements, together with the decision to transfer to Nature Communications, we now believe that this work is suitable for publication.

We thank the reviewer for helping us improve our manuscript and for recommending publication in *Nature*

Communications. The reviewer has no more suggestions or comments for us to address.

Reviewer #4 (Remarks to the Author: Overall significance):

The authors have fully addressed my concerns. I do not have further revisions to suggest at this moment.

We appreciate the reviewer's previous comments and suggestions, which helped us improve our manuscript. We are grateful for the recommendation of publication in *Nature Communications*. The reviewer has no more suggestions or comments for us to address.